Analysis

# The DNA-PAINT palette: a comprehensive performance analysis of fluorescent dyes

Philipp R. Steen [1,2], Eduard M. Unterauer[1,2], Luciano A. Masullo [2], Jisoo Kwon[2], Ana Perovic[2], Kristina Jevdokimenko[3], Felipe Opazo [3,4,5], Eugenio F. Fornasiero [3,6] & Ralf Jungmann [1,2] ✉

DNA points accumulation for imaging in nanoscale topography (DNA-PAINT) is a super-resolution fluorescence microscopy technique that achieves single-molecule 'blinking' by transient DNA hybridization. Despite blinking kinetics being largely independent of fluorescent dye choice, the dye employed substantially affects measurement quality. Thus far, there has been no systematic overview of dye performance for DNA-PAINT. Here we defined four key parameters characterizing performance: brightness, signal-to-background ratio, DNA-PAINT docking site damage and off-target signal. We then analyzed 18 fluorescent dyes in three spectral regions and examined them both in DNA origami nanostructures, establishing a reference standard, and in a cellular environment, targeting the nuclear pore complex protein Nup96. Finally, having identified several well-performing dyes for each excitation wavelength, we conducted simultaneous three-color DNA-PAINT combined with Exchange-PAINT to image six protein targets in neurons at ~16 nm resolution in less than 2 h. We thus provide guidelines for DNA-PAINT dye selection and evaluation and an overview of performances of commonly used dyes.

Super-resolution microscopy[1–6] has provided new insights into the life sciences by circumventing the classical diffraction limit of light. Notably, recent technological advances routinely achieve molecular-scale resolution[7,8]. Among the various methodologies employed in super-resolution microscopy, single-molecule localization microscopy (SMLM) is widely used. In SMLM, fluorescence emission is temporally and spatially separated by stochastic activation of target-bound fluorescent dyes. One of the most important factors in SMLM is achieving the highest spatial resolution, which is directly linked to obtaining high localization precisions. In conventional SMLM, high localization precisions fundamentally scale with the square root of the number of photons detected ($N$); thus, a high molecular brightness of the employed dyes is important. However, further dye properties, which directly depend on the specific SMLM implementation, are critical to consider.

Stochastic optical reconstruction microscopy (STORM)[4,9], for instance, requires photoswitchable organic dyes with appropriate switching kinetics. These requirements for STORM have been comprehensively laid out in a previous study[10].

DNA points accumulation for imaging in nanoscale topography (DNA-PAINT), another implementation of SMLM, uses the transient hybridization of dye-labeled 'imager' oligonucleotides to their complementary, target-bound 'docking' strands to achieve the necessary blinking for SMLM. However, this specific method of achieving single-molecule blinking leads to distinct requirements for fluorescent dyes. Photoswitching properties, for instance, are less important since blinking is governed by transient DNA hybridization. Recent advances in DNA-PAINT[8,11,12] have largely focused on sequence and acquisition assay optimization to improve DNA hybridization kinetics, while

[1]Faculty of Physics and Center for Nanoscience, Ludwig Maximilian University, Munich, Germany. [2]Max Planck Institute of Biochemistry, Martinsried, Germany. [3]Institute of Neuro- and Sensory Physiology, University Medical Center Göttingen, Göttingen, Germany. [4]Center for Biostructural Imaging of Neurodegeneration, University Medical Center Göttingen, Göttingen, Germany. [5]NanoTag Biotechnologies GmbH, Göttingen, Germany. [6]Department of Life Sciences, University of Trieste, Trieste, Italy. ✉e-mail: jungmann@biochem.mpg.de

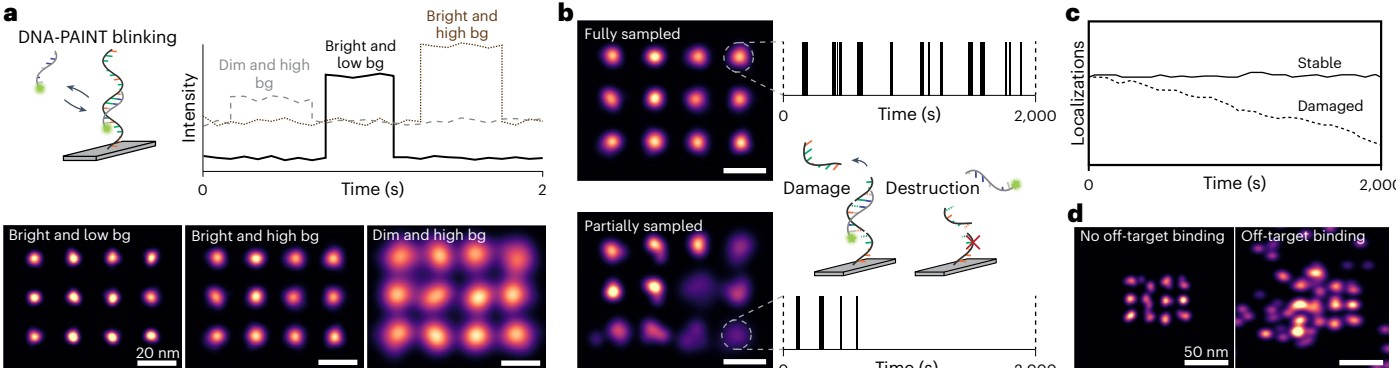

**Fig. 1 | Dye-dependent factors influencing DNA-PAINT image quality. a**, Top left: DNA-PAINT achieves blinking necessary for stochastic super-resolution microscopy via transient, repetitive DNA hybridization of a dye-labeled imager to its complementary target-bound docking strand. Top right: simulated intensity versus time traces representing three different SBRs influencing achievable localization precision and, thus, spatial resolution. Bottom: three simulated point grid patterns showcasing the respective effect on image quality and resolution. bg, background. Scale bars, 20 nm. **b**, Docking sites may be fully (top) or partially (bottom) sampled. The latter is indicative of photoinduced damage to docking sites, which may be dye dependent. Right: Localizations over time. Scale bars, 20 nm. **c**, This depletion of docking sites leads to a reduction of localizations over the course of the measurements and, thus, to a reduction in image quality. **d**, Potential off-target (that is, non-specific) binding of dye-labeled imager strands further influences DNA-PAINT image quality, which could be a dye- and sample-dependent effect. Scale bars, 50 nm.

optimal dye performance has not been systematically investigated so far. While it is broadly understood that the fluorescent dye Cy3B offers excellent performance in DNA-PAINT, there are instances in which neither the use of Cy3B for single-plex image acquisition nor Exchange-PAINT[13] (which uses orthogonal target DNA sequences and sequential image acquisition) are possible. The former applies when using samples tagged with red fluorescent protein or similar markers featuring spectra that overlap with that of Cy3B or imaging on microscope setups that do not feature a sufficiently powerful 560 nm laser line. The latter is the case when imaging samples such as dense tissues or nonadherent cells that risk being washed away during buffer exchange. Especially in these cases, spectral multiplexing without buffer exchange is an important technique to enable imaging of multiple targets.

Here, we introduce a comprehensive set of requirements to characterize dye performance for DNA-PAINT and apply them in vitro and in situ for 18 commonly used fluorescent dyes. We identified a set of best-performing dyes for 488 nm, 560 nm and 640 nm excitation (among which are, for example, CF488A, Cy3B and Atto643), which enabled us to combine sequential and spectral multiplexing to resolve six neuronal protein targets at 16 nm resolution in under 2 h.

## Results

### Factors determining performance

In general, the quality of DNA-PAINT measurements is determined by (1) the brightness and (2) signal-to-background ratio (SBR) of the dye, (3) potential photo-induced damage of docking strands and (4) the amount of off-target (unspecific) binding of imager strands (all depicted in Fig. 1).

Brightness and SBR are the main determining factors for localization precision, directly translating to spatial resolution, the ability to distinguish point objects. A large number of photons (>10,000) emitted per unit of time combined with a high SBR (>5) yields good localization precision (2 nm or better). A large number of photons with a low SBR yields a slightly reduced localization precision (approximately 3 nm), and a low number of photons, regardless of SBR, yields a 'poor' localization precision (5 nm or worse; Fig. 1a).

Damage to docking sites[14] does not directly affect the localization precision of individual blinking events, but the reduced sampling has negative consequences for overall performance (Fig. 1b,c). Primarily, DNA-PAINT-specific advantages, such as the ability to estimate the

number of docking sites in cases where they are positioned too close to each other to be resolved (qPAINT[15]) and resolution enhancement by sequential imaging (RESI[8]), are substantially impaired. Furthermore, the reduced sampling leads to a lower-quality image.

Finally, the proportion of off-target localizations (that is, 'sticking') affects image quality and quantitative analysis. In cases where single docking site resolution is not achieved, as in many cellular measurements, previously reported mechanisms of filtering unspecific localizations (which in turn require consistent sampling)[16] cannot be applied. Therefore, an ideal dye contributes as little as possible to off-target binding of the imager strand (Fig. 1d).

Eighteen dyes in the visible spectrum—blue-absorbing (Atto488, Alexa Fluor 488, Abberior Star 488 and CF488A), green-absorbing (Cy3B, Atto565, CF568 and Janelia Fluor 585) and red-absorbing (Cy5, Cy5B, Atto643, Atto647N, Atto655, CF640R, CF660R, Alexa Fluor 647, Janelia Fluor 646 and Abberior Star 635p)—were evaluated for performance in DNA-PAINT microscopy.

### DNA origami-based characterization

Quantitative comparisons of dye performances require a controlled and highly reproducible target system. DNA origami[17], which enables the nanometer-precise positioning of DNA strands, provides both controllability and reproducibility and is commonly used as a super-resolution reference structure[7,12,18,19]. For this study, we designed two DNA origami structures: one displaying twelve 20-nm-spaced docking strands (Fig. 2a), testing and illustrating achievable resolution, and the other featuring a single docking site (alongside barcoding sites for structure identification, see Extended Data Figs. 1 and 2) to extract parameters such as kinetics in a systematic manner. Barcodes and the dye of interest were imaged sequentially via Exchange-PAINT[13] with total internal reflection (TIR) illumination to ensure accurate identification of individual docking sites even when using poorly performing dyes (see Extended Data Fig. 1a–c and Methods for details).

Figure 2b shows the localizations detected for a well-performing dye, featuring consistent sampling over the duration of the measurement and high localization precision. DNA-PAINT images are rendered using Gaussians weighted by the localization precision as calculated according to Mortensen et al.[20] (Fig. 2c and Methods). A single widefield DNA-PAINT measurement with a 75 × 75 μm² field of view (FOV) generally captures more than 5,000 DNA origami structures (see Fig. 2d for a typical subregion).

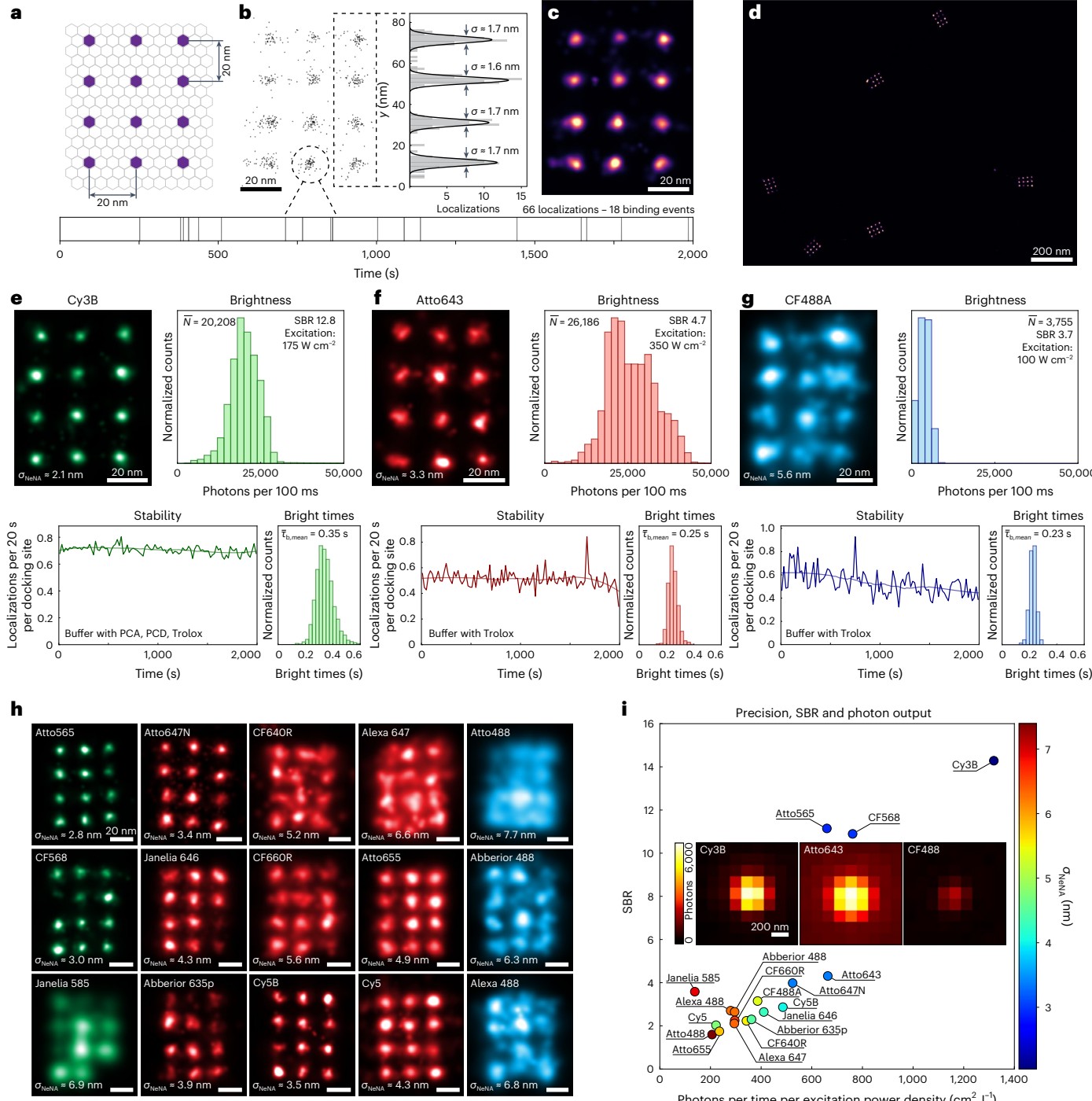

**Fig. 2 | Dye benchmarking using DNA origami structures. a**, DNA origami featuring 20-nm-spaced docking strands provides an exquisite reference structure for dye benchmarking. **b**, Individual localizations in an experiment are Gaussian distributed around the expected ground truth position of the docking strand with a certain standard deviation, a possible representation of localization precision (in this case, 1.7 nm). Each docking site is repeatedly sampled over the course of the measurement. **c**, Gaussian rendering of localizations weighted by their respective localization precisions yields a high-quality DNA-PAINT super-resolution image. **d**, Overview of multiple DNA origami, a subset of approximately 5,000 structures in a full 75 × 75 μm² FOV. **e–g**, Dye performance is primarily determined by achievable localization precision ($\sigma_{NeNA}$), which is

strongly dependent on brightness (mean number of photons, $\bar{N}$, per 100 ms) and SBR per localization. Further performance factors are stability (localizations per 20 s per docking site over time) and (to a lesser extent) mean bright times per binding event ($\bar{\tau}_b$). These factors are exemplified for Cy3B (**e**), Atto643 (**f**) and CF488A (**g**). **h**, Representative DNA origami structures for 15 further dyes alongside the localization precision achieved in the respective measurements. Scale bars, 20 nm. **i**, Combined performance evaluation for all dyes benchmarked in this study. SBR is plotted against normalized photon count per localization alongside color-coded average localization precision for each dye. Three repeats per dye. Insets: Representative single-molecule images for three dyes.

**Table 1 | Summary of DNA origami-based dye evaluation**

| Dye | Ex. maximum (nm) | Em. maximum (nm) | QY | Ext. (×10³ M⁻¹ cm⁻¹) | Buffer additive | Precision (nm) | Photons per 100 ms | Bright time (s) | SBR | Localization drop (%) | Destroyed binding sites (%) |
|---|---|---|---|---|---|---|---|---|---|---|---|
| Atto488 | 500 | 520 | 0.8 | 90 | Trolox | 7.4±0.3 | 2,073±249 | 0.27±0.02 | 1.6±0.2 | 44.3±2.8 | 17.1±3.6 |
| Alexa Fluor 488 | 499 | 520 | 0.92 | 73 | Trolox | 6.3±0.5 | 2,811±290 | 0.24±0.03 | 2.7±0.5 | 33.6±3.5 | 7.4±1.8 |
| Abberior Star 488 | 503 | 524 | 0.89 | 65 | Trolox | 6.3±0.6 | 2,969±365 | 0.21±0.02 | 2.7±0.7 | 27.6±4.5 | 9.1±2.7 |
| CF488A | 490 | 515 | – | 70 | Trolox | 5.4±0.6 | 3,879±732 | 0.20±0.03 | 3.2±0.8 | 33.7±10.4 | 8.3±3.6 |
| Cy3B | 560 | 571 | 0.58 | 120 | PCA PCD Trolox | 2.0±0.1 | 23,195±3,323 | 0.35±0.01 | 14.3±2.6 | 3.7±3.4 | 2.1±0.3 |
| Atto565 | 564 | 590 | 0.9 | 120 | Trolox | 2.9±0.1 | 11,600±1,896 | 0.24±0.01 | 11.1±2.8 | 16.4±4.0 | 1.3±0.5 |
| Janelia Fluor 585 | 585 | 609 | 0.78 | 156 | Trolox | 6.9±1.1 | 2,429±1,137 | 0.26±0.02 | 3.6±0.3 | 35.6±28.7 | 17.7±13.4 |
| CF568 | 562 | 583 | – | 100 | – | 3.0±0.2 | 13,388±2,033 | 0.21±0.01 | 10.9±2.5 | 11.5±1.1 | 2.3±0.2 |
| Cy5 | 651 | 670 | 0.27 | 250 | PCA PCD Trolox | 4.7±0.6 | 7,801±1,722 | 0.42±0.02 | 2.0±0.1 | 0.0±0.0 | 2.1±1.8 |
| Cy5B | 671 | 682 | 0.4 | 241 | PCA PCD Trolox | 4.0±0.4 | 17,090±2,765 | 0.25±0.02 | 2.9±0.5 | 0.3 CI (0.0–0.8) | 2.1±0.5 |
| Atto643 | 643 | 665 | 0.62 | 150 | Trolox | 3.4±0.3 | 23,327±6,241 | 0.25±0.01 | 4.3±0.7 | 12.1±4.5 | 1.9±0.5 |
| Atto647N | 646 | 664 | 0.65 | 150 | Trolox | 3.4±0.1 | 18,448±3,523 | 0.32±0.01 | 4.0±0.1 | 18.7±9.2 | 4.0±1.9 |
| Atto655 | 663 | 680 | 0.3 | 125 | – | 5.8±1.7 | 8,273±2,732 | 0.27±0.04 | 1.7±0.1 | 8.9±4.0 | 4.0±2.7 |
| CF640R | 642 | 662 | – | 105 | – | 5.5±1.0 | 12,024±3,240 | 0.22±0.01 | 2.2±0.2 | 7.5±2.5 | 2.7±1.2 |
| CF660R | 663 | 682 | – | 100 | – | 6.6±0.9 | 10,399±1,309 | 0.21±0.02 | 2.3±0.3 | 4.2±3.8 | 2.8±0.8 |
| Abberior Star 635p | 638 | 651 | 0.9 | 120 | Trolox | 4.5±0.7 | 12,731±1,037 | 0.23±0.01 | 2.3±0.3 | 53.3±24.8 | 17.9±11.6 |
| Janelia Fluor 646 | 646 | 664 | 0.54 | 152 | – | 4.3±0.1 | 14,440±3,596 | 0.25±0.03 | 2.6±0.4 | 23.3±9.1 | 5.6±2.6 |
| Alexa Fluor 647 | 650 | 671 | 0.33 | 270 | PCA PCD Trolox | 6.3±0.2 | 10,348±1,358 | 0.19±0.01 | 2.1±0.2 | 1.4 CI (0.0–2.9) | 1.2±0.1 |

Excitation (Ex.) maxima, emission (Em.) maxima, quantum yields (QY) and extinction coefficients (Ext.) from respective manufacturers. QY were not available for CF dyes. Performance for optimal buffer additive is shown. Uncertainties represent s.d. and CI describes 68% confidence interval from $N = 3$ repeat measurements.

The average localization precision $\sigma_{NeNA}$ for each measurement was estimated using the nearest-neighbor-based metric (NeNA[21]). Photon count, bright times, SBR, localizations over time and the percentage of destroyed docking sites were extracted from single docking site data as described in Methods. A series of successive localizations (each by definition one frame long) is interpreted as a binding event of an imager strand to the respective docking site. The first and last localizations of a binding event exhibit a lower photon count than the rest of the localizations due to the imagers binding or unbinding during the frame exposure time. To accurately determine the average photon output of each dye over time, only localizations that are not the first or last of a binding event were considered (Extended Data Fig. 1d–f). Bright times and SBR were calculated as defined in Methods (see Extended Data Fig. 3 for details on photon count and SBR calculation). The average number of localizations over time for all docking strands in a measurement, the reduction of which is correlated with damage to the docking strands (Extended Data Fig. 4), were plotted. Since the docking sequences used are concatenated[12] (they feature multiple sites for imagers to bind to), partial damage to a docking site will reduce the effective imager on-rate while not outright destroying the docking site. Finally, the percentage of destroyed docking sites was estimated by examining the time between the last recorded binding event and the end of the measurement (see Methods and Extended Data Fig. 5 for an overview of this process). The numerical results of these parameters for all dyes examined are presented in Table 1. The detailed visual results for three representative dyes, one for each wavelength, are shown in Fig. 2e–g. The localization precision and, thus, spatial resolution for each remaining dye are visually represented in Fig. 2h. Notably, each excitation wavelength features multiple dyes with similar performances: Cy3B, Atto565 and CF568 for green; Atto643, Atto647N and Cy5B for red; and CF488A, Abberior Star 488 and Alexa Fluor 488 for blue excitation.

The photoinduced damage to DNA docking sites is dependent on dye choice, laser power and laser wavelength. Therefore, the laser power chosen is always a compromise between photon output (high laser powers) and low damage (low laser powers). Shorter wavelengths exhibit substantially higher damage to docking sites. Excitation intensities (100 W cm⁻² for 488 nm excitation, 175 W cm⁻² for 560 nm excitation and 350 W cm⁻² for 640 nm excitation; see Methods for power density calculation) were thus chosen to minimize damage while yielding high photon output and, therefore, resolution (Extended Data Fig. 6). A key advantage of DNA-PAINT is the consistent, repetitive sampling of a given docking site over the course of the measurement. This not only enables quantitative analyses such as qPAINT[15] but also facilitates distinguishing imager sticking events (off-target localizations) from true docking sites on the basis of kinetic fingerprinting. A substantial decrease in the number of localizations over time or the outright destruction of docking sites severely limits the applicability of these techniques, hampering data analysis and downstream interpretation of results. In general, we recommend using the lowest excitation power that ensures resolving the target of interest.

Various buffer additives are commonly used in SMLM to reduce adverse effects such as transient dark states, dye bleaching or reactive oxygen species generation[22]. In DNA-PAINT, typically an enzymatic

oxygen scavenging system (3,4-dihydroxybenzoic acid (PCA) and protocatechuate 3,4-dioxygenase from *Pseudomonas* (PCD)[23]) alongside a triplet-state quencher ((+/−)-6-hydroxy-2,5,7,8-tetra-methylchromane-2-carboxylic acid, Trolox[24]) is used[18]. To evaluate the influence of these additives, each dye was imaged using no additive, only Trolox and PCA/PCD/Trolox. The results for all experiments are given in Supplementary Figs. 1–18. For further experiments, including those shown in Fig. 2, only the best-performing buffer composition (indicated in Table 1) was used. The achievable localization precision is primarily governed by high photon output and SBR (Fig. 2i); however, docking site damage reduces overall image quality.

The other central component of a DNA-PAINT imager, apart from the dye, is the DNA sequence. Detailed investigations of optimal sequences, focusing on hybridization kinetics, have been previously performed[11,12]. Here, we further investigated whether the chosen imager sequence has any substantial effect on key performance metrics for three fluorescent dyes: Cy3B, Atto643 and CF488A. The results, shown in Extended Data Fig. 7, reveal no notable differences between the sequences apart from slightly higher photon counts as well as longer bright times for sequence R2, consistent with previously reported values[12].

## Characterization in cells

Although DNA origami structures represent an ideal platform to investigate achievable localization precision and docking site damage in DNA-PAINT, this does not necessarily translate directly to a cellular system. Most notably, the complex cellular environment poses additional challenges such as increased potential for unspecific interaction of dye-labeled imager strands with cellular components. Thus, dye performance in a cellular environment is a critical measure for suitability in biological applications.

We chose to image Nup96, a structural component of the nuclear pore complex (NPC), as it represents a well-established reference structure for super-resolution microscopy[25,26]. The NPC is a macromolecular structure with a well-defined and conserved geometry, consisting of two rings, one nuclear and one cytoplasmic, each containing an eightfold radially symmetrical arrangement of Nup96 pairs yielding a total of 32 copies (Fig. 3a)[27]. Each cell was imaged three-dimensionally using highly inclined and laminated optical sheet (HILO) illumination (Fig. 3b), first using Cy3B with PCA/PCD/Trolox. This image was used to verify good labeling efficiency, correct sample position, focus and all other non-dye-related parameters that could influence measurement performance. Next, the same cell was imaged using the dye of interest. To ensure that neither the first imaging round nor the process of imager exchange alters or damages the sample, which would lead to biased results, the process was validated using Cy3B under identical conditions for both imaging rounds. The photon counts, SBR, localization precision, sampling rate and bright times were consistent for both imaging rounds.

Figure 3c–e illustrates well-performing dyes (Cy3B, Atto643 and CF488A) for three excitation wavelengths. The dye performances (Fig. 3f) are consistent with the DNA origami results: multiple dyes for each excitation wavelength exhibit similar performances.

Another factor for cellular applications is the dye specificity, that is, low off-target binding ('sticking'). We define 'dye specificity in an individual measurement', $S_{Dye}$, as the ratio of localizations per area originating from the NPC over the localizations per area in the cytoplasm (see Fig. 3g–i for representative regions). This, however, is not suitable for comparing dye performances owing to sample heterogeneity inherent in biological samples: labels (for example, DNA-labeled nanobodies) bound outside the nucleus (either on or off-target), for instance, would be incorrectly classified as dye sticking. Therefore, we introduce the measure of 'relative specificity', $R_{Dye}$, as the ratio of $S_{Dye}$ (dye specificity in an individual measurement for the dye of interest) over $S_{Cy3B}$ (dye specificity in an individual measurement for the Cy3B reference round). Critically, the areas compared are identical in both cases (see

Extended Data Fig. 8 for an illustration of this process). Therefore, sample heterogeneity is of no concern. The relative specificity $R_{Dye}$ for all dyes compared with Cy3B is shown in Fig. 3j. Interestingly, we observe no correlation between relative specificity and localization precision.

Methods to filter out localizations originating from unspecific dye sticking have been previously reported[8,16]. These, however, require single-protein resolution as a prerequisite for filtering. Thus, a superior resolution (Cy3B) is arguably of greater importance than reduced off-target binding—unless the target cannot be resolved, in which case it would be preferable to employ a lower-resolution yet higher-specificity dye such as CF640R or Atto655.

Finally, we explored the consistency of dye metrics obtained through Nup96 imaging in comparison with another cellular target, specifically the mitochondrial membrane-associated protein Tom20. To this end, U2OS cells were stained with primary anti-Tom20 antibodies in combination with secondary, R1-docking site labeled nanobodies. A subset (Cy3B, CF568, Janelia 646, Atto643 and CF488A) of dyes were then evaluated (Extended Data Fig. 9). For Cy3B, the achieved resolution is equivalent to current state-of-the-art measurements[28]. Overall, resolutions and relative specificities of the dyes tested show consistent trends with those measured in NPCs, indicating that our performance metrics apply broadly to cellular targets.

## Spectral multiplexing in neurons

With multiple well-performing dyes for each excitation wavelength at our disposal, we can now conduct simultaneous three-color spectral multiplexing in addition to conventional Exchange-PAINT. This enables three-times-faster acquisition and reduces the time needed for buffer exchanges in Exchange-PAINT. We chose CF488A, Cy3B and Atto643 as they all perform well with the buffer additive Trolox and achieve high resolutions for their respective excitation wavelengths. We modified a standard inverted fluorescence microscope by adding relay lenses, dichroic mirrors and three scientific complementary metal oxide-semiconductor (sCMOS) cameras (see Extended Data Fig. 10 for the design). The system was designed so that cameras 1–3 record the emission from CF488A, Cy3B and Atto643, respectively. An additional camera in a separate imaging path was used for alignment and performance verification. To quantify cross-talk in terms of localizations, we imaged DNA origami with the three dyes sequentially and compared the number of detected localizations for each camera. Cross-talk (the ratio of the number of localizations detected in the 'incorrect' channel over the number of localizations detected in the 'correct' channel) was below 1% for all cases. To demonstrate the capabilities of three-color simultaneous imaging, we chose to image neurons. Since six speed-optimized DNA-PAINT imager sequences are currently available[12], we combined spectral multiplexing with a single round of Exchange-PAINT and one-step immunofluorescence labeling using secondary nanobodies[29] to acquire six targets in less than 2 h, including the time needed for buffer exchange. Targets were chosen to illustrate neuronal and synaptic architecture (βII-spectrin, neurofilament and bassoon), differentiate synapses (VGAT-1 and PSD-95) and provide cellular context (Tom20).

Figure 4a and Fig. 4b illustrate the three 67 × 67 μm² FOVs acquired in imaging rounds 1 and 2, respectively. The achieved localization precisions (~7 nm) are comparable to sequential Cy3B imaging of the same targets[30]. Overlaying all six channels (Fig. 4c) illustrates neuronal architecture and context at state-of-the-art spatial resolutions. Importantly, excitatory synapses (Fig. 4d,e) can be clearly identified and differentiated from inhibitory synapses (Fig. 4f,g). Interestingly, the inhibitory synapses frequently colocalize with mitochondria (marked by Tom20), suggesting high energy consumption of inhibitory synapses. The characteristic 190-nm-spaced spectrin rings[31] were clearly resolved using CF488A (Fig. 4h). Thicker bundles and individual fibers of neurofilaments were resolved (Fig. 4i) and individual Tom20 proteins on mitochondria were visible at approximately 14 nm resolution (Fig. 4j).

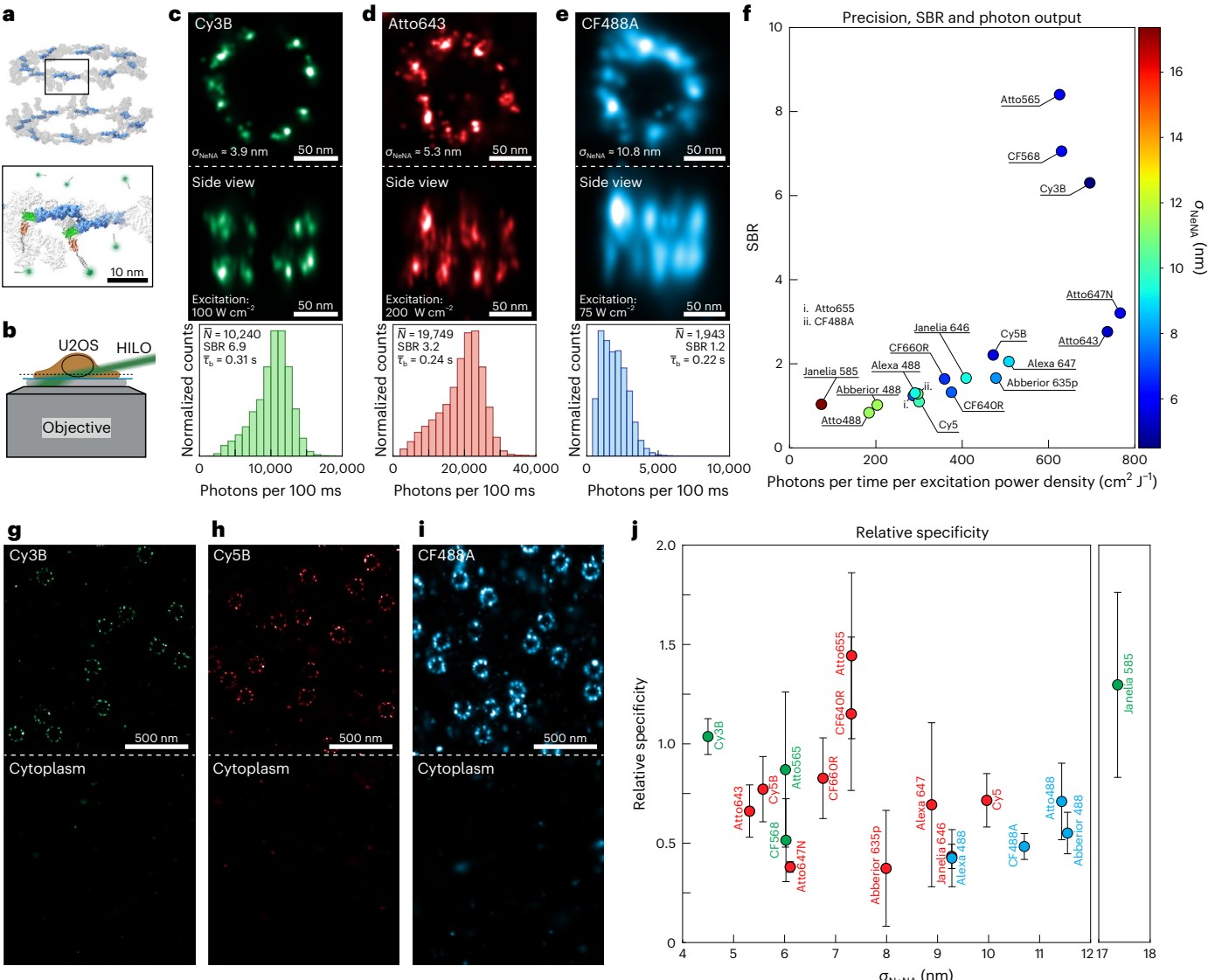

**Fig. 3 | Dye benchmarking in cells. a**, Nup96-mEGFP in U2OS cells is labeled using DNA-conjugated anti-GFP nanobodies. The structure was adapted from PDB 7PEQ (ref. 27). Box indicates expansion. **b**, 3D imaging is conducted using HILO illumination and an astigmatic lens in the detection path (not shown). **c**, Cy3B three-dimensionally resolves the overall NPC structure as well as nuclear and cytoplasmic rings. **d**, Atto643, despite a higher photon count, exhibits a reduced localization precision compared with Cy3B, yielding a lower image quality. **e**, CF488 reveals the planar ring structure but struggles to resolve the nuclear and cytoplasmic rings three-dimensionally. **f**, Photon count, SBR and localization precision for all examined dyes (3 repeats per dye) exhibit similar

trends to Fig. 2i. **g**–**i**, Comparison of nuclear (top) versus cytoplasmic (bottom) localizations enables the quantification of dye-dependent off-target (nonspecific) interactions. Representative regions shown for Cy3B (**g**), Cy5B (**h**) and CF488A (**i**). **j**, Using Cy3B as a reference for each cell and dye, the relative specificity (that is, whether a dye exhibits less or more off-target localizations than Cy3B) is plotted for all examined dyes. Except for CF640R, Atto655 and Janelia Fluor 585, all dyes exhibit increased off-target binding compared with Cy3B. The circles represent the mean relative specificity, and the error bars represent the s.d. across 3 repeats per dye.

The high multiplexing speed suggests the viability of this technique for rapid screening of clinically relevant neuron samples, enabling early identification of pathologies. Furthermore, all previously reported mechanisms of increasing DNA-PAINT acquisition speed can be combined with spectral multiplexing as presented here, increasing multiplexing capabilities (and thereby dividing acquisition time) by a factor of 3. This, in turn, furthers the viability of DNA-PAINT for high-throughput screens or large-area imaging.

## Discussion

The choice of fluorescent dye is critical to obtain high performance in DNA-PAINT experiments. Here, we demonstrated that the performance of fluorescent dyes is characterized by brightness, SBR, docking site damage and the degree of off-target binding. We analyzed 18

fluorescent dyes for these parameters in DNA origami and in cells and found optimal candidates for three excitation wavelengths. For 488 nm excitation, which exhibits the lowest performance of the wavelengths investigated, CF488A performs the best and achieves precisions around 10 nm in cells. For 560 nm, Cy3B and for 640 nm excitation, Atto643, Atto647N and Cy5B perform best, achieving precisions of 4–5 nm in cells. Furthermore, we screened different buffer additives to determine the ideal system for each dye: Trolox for CF488A, Atto643 and Atto647N and PCA, PCD and Trolox for Cy3B and Cy5B.

We conducted this dye screening using a standard fluorescence microscope with standard laser lines (488 nm, 560 nm and 640 nm) and commonly used dichroics, thus best representing the conditions found in most laboratories.

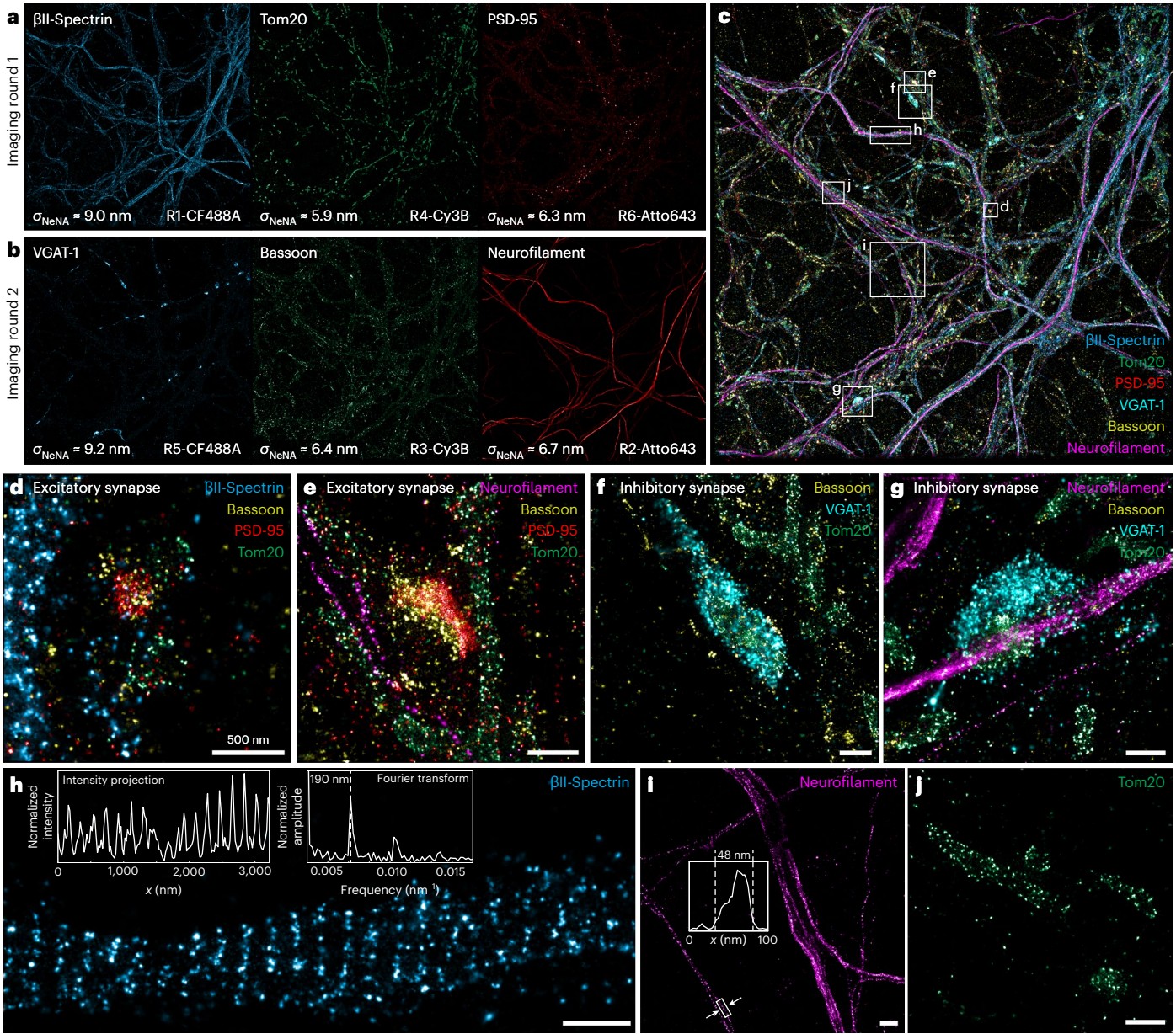

**Fig. 4 | Spectral multiplexing in neurons. a**, βII-Spectrin, Tom20 and PSD-95 were imaged simultaneously using CF488A, Cy3B and Atto643, respectively. R1–R6 designate DNA-PAINT imager sequences (see Supplementary Table 2). **b**, After one round of Exchange-PAINT, VGAT-1, bassoon and neurofilament were imaged simultaneously. **c**, Overlay of the six targets acquired in 100 min of imaging. **d,e**, Excitatory synapses are clearly identified by the presence of bassoon and PSD-95. βII-spectrin (**d**) and neurofilament (**e**) reveal structural context. **f,g**, Inhibitory synapses are characterized by the presence of VGAT-1.

The synapses shown here contain mitochondria, marked by Tom20, indicative of high energy consumption. Neurofilament (**g**) indicates structural context. **h**, Characteristic βII-Spectrin rings with a periodicity of 190 nm (see insets) are clearly evident. **i**, Neurofilament is present in single filaments (see inset) as well as thicker bundles of fibers. Inset: Intensity projection of the indicated region. **j**, Labeled Tom20 proteins on the mitochondria are individually resolved. Image size in **a**–**c** (full FOV) is 67 × 67 μm². Scale bars, 500 nm. Imaging was repeated three times with similar results; one dataset is shown.

We are confident that the criteria and analysis processes defined here can become a benchmark for future systematic investigations of DNA-PAINT performance, including different dyes, different excitation and detection regimes, and more buffer additives. The well-defined structures of DNA origami and the NPC ensure consistency and comparability between tested conditions, and the clear, measurable performance parameters enable direct comparison of results.

Finally, the spectral distinction of 488 nm-, 560 nm- and 640 nm-excited fluorophores together with the availability of well-performing fluorophores for each wavelength enabled simultaneous multichannel detection, which we applied to increase Exchange-PAINT's acquisition speed by a factor of 3 in neurons, further making DNA-PAINT a viable tool for high-throughput screens.

## Online content

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

## Methods

### Materials

Unmodified DNA oligonucleotides were purchased from Integrated DNA Technologies. DNA oligonucleotides modified with Atto488, Abberior Star 488, Cy3B, Janelia Fluor 585, Atto565, Atto647N, Atto655, Abberior Star 635p and Janelia Fluor 646 were obtained from Metabion. DNA oligonucleotides modified with Alexa Fluor 488 were obtained from Integrated DNA Technologies. Eurofins Genomics supplied DNA oligonucleotides modified with Atto643, Cy5 and Alexa Fluor 647. Succinimidyl esters of CF488A, CF565, CF640R and CF660R were ordered at biotium. Cy5B[32] was kindly gifted by the Schnermann and Tinnefeld labs. M13mp18 scaffold was obtained from Tilibit. Magnesium 1 M (cat. no. AM9530G), sodium chloride 5 M (cat. no. AM9759), ultrapure water (cat. no. 10977-035), Tris 1 M pH 8 (cat. no. AM9855G), EDTA 0.5 M pH 8.0 (cat. no. AM9260G) and 10× phosphate-buffered saline (PBS; cat. no. 70011051) were purchased from Thermo Fisher Scientific. Bovine serum albumin (BSA; cat. no. A4503-10G) was ordered from Sigma-Aldrich. Triton X-100 (cat. no. 6683.1), sodium borohydride >97% (cat. no. 4051.1), ammonium chloride (cat. no. K298.1) and potassium chloride (cat. no. 6781.1) were purchased from Carl Roth. Sodium hydroxide (cat. no. 31627.290) was purchased from VWR. Paraformaldehyde (cat. no. 15710) and glutaraldehyde (cat. no. 16220) were obtained from Electron Microscopy Sciences. Tween-20 (cat. no. P9416-50ML), glycerol (cat. no. 65516-500ml), methanol (cat. no. 32213-2.5L), PCD (cat. no. P8279), PCA (cat. no. 37580-25G-F) and Trolox (cat. no. 238813-5G) were ordered from Sigma-Aldrich. Neutravidin (cat. no. 31000) was purchased from Thermo Fisher. BSA-biotin (cat. no. A8549) was obtained from Sigma-Aldrich. Coverslips (cat. no. 0107032) and glass slides (cat. no. 10756991) were purchased from Marienfeld and Thermo Fisher Scientific. Fetal bovine serum (cat. no. 10500-064), 1× PBS pH 7.2 (cat. no. 20012-019) and 0.05% trypsin–EDTA (cat. no. 25300-054) were purchased from Thermo Fisher Scientific, and 90-nm-diameter gold nanoparticles (cat. no. G-90-100) were ordered from Cytodiagnostics.

### Buffers

- Buffer A: 10 mM Tris-HCl pH 8, 100 mM NaCl and 0.05% Tween-20; pH 8
- Buffer B: 10 mM MgCl$_2$, 5 mM Tris-HCl pH 8, 1 mM EDTA and 0.05% Tween-20; pH 8
- Buffer C: 1× PBS, 0.1 mM EDTA, 500 mM NaCl and 0.05% Tween; pH 7.4
- Folding buffer: 10 mM Tris, 1 mM EDTA and 12.5 mM MgCl$_2$; pH 8
- FoB5 buffer: 5 mM Tris, 1 mM EDTA, 5 mM NaCl and 5 mM MgCl$_2$; pH 8
- Blocking buffer: 1× PBS, 1 mM EDTA, 0.02% Tween-20, 0.05% NaN$_3$, 2% BSA and 0.05 mg ml$^{-1}$ sheared salmon sperm DNA
- PCD buffer: 100 mM Tris-HCl pH 8, 50 mM KCl, 1 mM EDTA and 50% glycerol.

### PCA, PCD and Trolox

The 40× PCA was prepared by mixing 154 mg PCA in 10 ml water and NaOH and adjusting the pH to 9.0. The 100× PCD was prepared by adding 2.3 mg PCD to 3.3 ml of PCD buffer. The 100× Trolox was prepared by first adding 100 mg Trolox to 430 µl of 100% methanol and 3.2 ml water, then adding 350 µl of 1 M NaOH and finally adding ~480 µl of 1 M NaOH in 40 µl increments, shaking the solution each time, until the Trolox was completely dissolved.

### Dye–DNA conjugation

CF488A, CF565, CF640R, CF660R and Cy5B were conjugated as follows: DNA oligonucleotides were reacted with fourfold molar excess of NHS-modified dyes in borate buffer (pH 8.5) for 1 h at 25 °C and 300 rpm on a shaker. Unconjugated DNA and dye were removed by anion exchange chromatography using a liquid chromatography system (ÄKTA pure, GE Healthcare) equipped with a Resource Q 1 ml column (Cytiva).

### Nanobody–DNA conjugation

Nanobodies against green fluorescent protein (GFP) were purchased from NanoTag Biotechnologies (cat. no. N0305) with a single ectopic cysteine at the C-terminus for site-specific and quantitative conjugation. The conjugation to 5xR1 DNA-PAINT docking sites was performed as described previously[33]. First, buffer was exchanged to 1× PBS + 5 mM EDTA pH 7.0 using Amicon centrifugal filters (10,000 Da molecular weight cutoff). Free cysteines were reacted with 20-fold molar excess of bifunctional dibenzocyclooctyne (DBCO)-maleimide linker (Sigma-Aldrich, cat. no. 760668) for 2–3 h on ice. Unreacted linker was removed by buffer exchange to PBS using Amicon centrifugal filters. Azide-functionalized DNA was added with 5–10 molar excess to the DBCO nanobody and reacted overnight at 4 °C. Unconjugated nanobody and free azide DNA was removed by anion exchange using an ÄKTA pure liquid chromatography system equipped with a Resource Q 1 ml column.

### DNA origami self-assembly

All DNA origami structures were designed in Picasso Design[18]. Self-assembly of DNA origami was accomplished in a one-pot reaction mix with a total volume of 50 µl, consisting of 10 nM scaffold strand (for sequence, see Supplementary Table 5), 100 nM folding staples (Supplementary Data 1), 500 nM biotinylated staples (Supplementary Data 1) and 1 µM staple strands with docking site extensions (Supplementary Data 1) in folding buffer. The reaction mix was then subjected to a thermal annealing ramp using a thermocycler. First, it was incubated at 80 °C for 5 min, cooled using a temperature gradient from 60 °C to 4 °C in steps of 1 °C per 3.21 min and finally held at 4 °C. DNA origami with twelve 20-nm-spaced R1, R2 or R6 docking sites as well as DNA origami with a single R1, R2 or R6 docking site alongside R3 barcode docking sites were folded (Extended Data Fig. 2).

### DNA origami purification

DNA origami structures were purified via ultrafiltration using Amicon Ultra Centrifugal Filters with a 100 kDa molecular weight cutoff (Merck Millipore, UFC510096) as previously described[34]. Folded origami was brought to 500 µl with FoB5 buffer and spun for 3 min 30 s at 10,000g. This process was repeated twice. Purified DNA origami structures were recovered into a new tube by centrifugation for 5 min at 5,000g. Purified DNA origami were stored at −20 °C in DNA LoBind tubes (Eppendorf, 0030108035).

### DNA origami sample preparation

For sample preparation, a bottomless six-channel slide (ibidi, 80608) was attached to a coverslip. First, 200 µl of biotin-labeled BSA (1 mg ml$^{-1}$, dissolved in buffer A) was flushed into the chamber and incubated for 2 min. The chamber was then washed with 200 µl of buffer A. Then, 200 µl neutravidin (0.1 mg ml$^{-1}$, dissolved in buffer A) was flushed into the chamber and allowed to bind for 2 min. After washing with 200 µl of buffer A and subsequently with 200 µl of buffer B, 60 µl of biotin-labeled DNA structures (~200 pM, equal parts 20 nm grid structures and barcoded single docking site structures) in buffer B was flushed into the chamber and incubated for 6 min. After DNA origami incubation, the chamber was washed 3× with 200 µl of buffer B.

### DNA origami imaging

Two-hundred microliters of the imager solution in buffer B was flushed into the chamber. Between imaging rounds, the sample was washed with 3–4 ml buffer B until no residual signal from the previous imager solution was detected. Then, the next imager solution was introduced. Imaging rounds were conducted with R1 imagers[12], barcode identification rounds with R1 Cy3b and R3 Cy3b with PCA, PCD and Trolox (Supplementary Table 1). All docking and imager sequences are listed in

Supplementary Table 2. The barcode identification rounds were always performed first, as imaging with Cy3B and PCA/PCD/Trolox causes virtually no damage to docking sites. This ensured accurate identification even if sites were destroyed during the rounds imaging the dyes of interest. For the sequence investigations, R1 was replaced by R2 or R6.

## U2OS cell culture
U2OS-CRISPR-Nup96-mEGFP cells (a gift from the Ries and Ellenberg laboratories) were cultured in McCoy's 5A medium (Thermo Fisher Scientific, 16600082) supplemented with 10% fetal bovine serum. Cells were passaged every 2–3 days using trypsin–EDTA.

## Nup96 EGFP sample preparation
U2OS-CRISPR-Nup96-mEGFP cells were seeded on eight-well high glass-bottom chambers (ibidi, 80807) at a density of 30,000 cells per 1 cm$^2$. Cells were fixed with 2.4% paraformaldehyde in PBS for 30 min at room temperature. After fixation, cells were washed three times with PBS. Blocking and permeabilization was performed with 0.25% Triton X-100 in blocking buffer for 90 min. After washing with PBS, cells were incubated overnight in blocking buffer at 4 °C with 100 nM R1 docking strand-coupled anti-GFP nanobodies. Unbound nanobodies were removed by washing three times with PBS, followed by washing once with buffer C for 10 min. Gold nanoparticles (1:3 dilution in PBS) were incubated for 5 min and washed three times with PBS. Postfixation was performed with 2.4% paraformaldehyde in PBS for 7 min. Finally, the wells were washed 3× with PBS.

## Nup96 EGFP imaging
Imager solution in buffer C was flushed into the chamber. Between imaging rounds, the sample was washed with 3–4 ml of PBS until no residual signal from the previous imager solution was detected. Then, the next imager solution was introduced. Imaging rounds were conducted with R1 imagers, identification and reference with R1 Cy3b with PCA, PCD and Trolox (Supplementary Table 1).

## Tom20 sample preparation
U2OS-CRISPR-Nup96-mEGFP cells were seeded on eight-well high glass-bottom chambers (ibidi, 80807) at a density of 30,000 cells per 1 cm$^2$. Cells were fixed with 4% paraformaldehyde in PBS for 15 min at room temperature. After fixation, cells were washed three times with PBS. Permeabilization was performed with 0.2% Triton X-100 in PBS for 30 min. After washing three times with PBS, blocking was performed using 3% BSA in PBS with 0.05 mg ml$^{-1}$ sheared salmon sperm DNA for 45 min. Cells were washed three times with PBS, then gold nanoparticles (1:3 dilution in PBS) were incubated for 5 min and washed three times with PBS. Cells were incubated overnight with 1.5 μl primary anti-Tom20 antibodies (Abcam, ab186735) and 1 μl R1 docking strand-coupled anti-rabbit nanobodies (NanoTag, N2405) in 300 μl blocking buffer. Unbound reagents were removed by washing four times with PBS and once with buffer C.

## Tom20 imaging
Imager solution in buffer C was flushed into the chamber. Between imaging rounds, the sample was washed with 3–4 ml of PBS until no residual signal from the previous imager solution was detected. Then, the next imager solution was introduced. Imaging rounds were conducted with R1 imagers.

## Animals
Wild-type Wistar rat pregnant mothers or pups (*Rattus norvegicus*) were obtained from the University Medical Center Göttingen and were handled according to the specifications of the University of Göttingen and of the local authority, the State of Lower Saxony (Landesamt für Verbraucherschutz, LAVES, Braunschweig, Germany). Animal experiments were approved by the local authority, the Lower Saxony State Office for Consumer Protection and Food Safety (Niedersächsisches Landesamt für Verbraucherschutz und Lebensmittelsicherheit).

## Neuron primary cell culture
Primary hippocampal neuron cultures from postnatal day 2 Wistar rat pups were prepared as described previously[35]. Briefly, upon dissection, hippocampal neurons were seeded onto 18 mm-diameter coverslips at a density of 60,000 cells per coverslip. Two hours after plating, once the cells attached to the coverslip, the medium was removed and fresh B27-supplemented neurobasal medium was supplied. Neurons were grown until 14 days in vitro. Following this culture method, the neurons developed proper polarity, generated intricate axonal and dendritic networks, and established multiple functional synaptic connections with each other[36].

## Six-plex neuron imaging
Rat primary hippocampal neurons were fixed using 4% paraformaldehyde for 30 min at room temperature, washed four times with PBS. After fixation, neurons were quenched using 100 mM NH$_4$Cl (Merck, 12125-02-9) in PBS. Permeabilization was done with 0.1% Triton X-100 (Sigma-Aldrich, 9036-19-5, X100-500ml) in PBS for 20 min. For blocking, 3% BSA in PBS was supplemented with 0.05 mg ml$^{-1}$ sheared salmon sperm DNA and incubated onto the sample for 45 min. Afterwards, the samples were washed with PBS, and gold nanoparticles (1:3 dilution in PBS) were incubated for 5 min and subsequently used as fiducial markers. Antibodies and their respective secondary nanobody partners (Supplementary Table 3) were preincubated in 10 μl blocking buffer at room temperature overnight. After preincubation, an excess (molar ratio of 1:2) of unlabeled secondary nanobody was introduced for 5 min (NanoTag Biotechnologies cat. nos. K0102-50 and K0202-50). The preincubated primary antibody and secondary nanobody complexes were pooled in 300 μl blocking buffer and added to the sample for 90 min. Then, the sample was washed five times with PBS and once with buffer C, and the imager solution for the first set of three targets was applied. After acquisition, the sample was washed three times with buffer C and the imager solution for the second set of three targets was applied (Supplementary Table 1 for imaging conditions).

## Microscope setup (single-color)
Fluorescence imaging for DNA origami and U2OS-Nup96 experiments was carried out on an inverted microscope (Nikon Instruments, Eclipse Ti2) with the Perfect Focus System, applying an objective-type TIRF configuration equipped with an oil-immersion objective (Nikon Instruments, Apo SR TIRF ×100, numerical aperture 1.49, oil). 488 nm, 560 nm and 642 nm lasers (MPB Communications, 1 W) were used for excitation and coupled into the microscope via a Nikon manual TIRF module. The laser beams were passed through cleanup filters (Chroma Technology, ZET488/10× for 488 nm excitation, ZET561/10× for 560 nm excitation and ZET642/20× for 642 nm excitation) and coupled into the microscope objective using a beam splitter (Chroma Technology, ZT488rdc-UF2 for 488 nm excitation, ZT561rdc-UF2 for 560 nm excitation and ZT647rdc-UF2 for 642 nm excitation). Fluorescence was spectrally filtered with an emission filter (Chroma Technology, ET525/50 m and ET500lp for 488 nm excitation, ET600/50 m and ET575lp for 560 nm excitation and ET705/72 m and ET665lp for 642 nm excitation) and imaged on an sCMOS camera (Hamamatsu, ORCA-Fusion BT) without further magnification, resulting in an effective pixel size of 130 nm (after 2 × 2 binning). The central 1,152 × 1,152 pixels (576 × 576 after binning) of the camera were used as the region of interest, and the scan mode was set to 'ultra quiet scan'. Three-dimensional (3D) imaging was performed using an astigmatism lens (Nikon Instruments, N-STORM) in the detection path[37]. Raw microscopy data were acquired using μManager[38] (Version 2.0.1). TIR illumination was used for the DNA origami data. HILO was employed for the acquisition of the NPC data.

## Microscope setup (three-color)

Fluorescence imaging for neurons was performed on an inverted microscope (Nikon Instruments, Eclipse Ti) with the Perfect Focus System, applying an objective-type TIRF configuration equipped with an oil-immersion objective (Nikon Instruments, Apo SR TIRF ×100, numerical aperture 1.49, oil). 488 nm (Cobolt, 200 mW), 560 nm and 642 nm lasers (MPB Communications, 1 W) were used for simultaneous excitation. The laser lines were combined using dichroic mirrors (Chroma Technology, ZT488rdc-UF2 and ZT561rdc-UF2), coupled into a single-mode optical fiber (Schäfter+Kirchhoff, PMC-E-530Si-4.0-NA009-3-APC.EC-150-P), collimated (Thorlabs, AC254-045-A-ML), expanded in a telescope (Thorlabs, AC254-050-A-ML and ACT508-200-A-ML) and focused (Thorlabs, ACT508-200-A-ML) on the back focal plane of the objective. TIRF positioning was achieved using three mirrors mounted on a one-axis translatable stage (TIRF stage; Extended Data Fig. 10). A quad-band filter cube (Chroma, TRF89901-EMv2-NK_Nikon TE2000 Laser TIRF Cube) without single-band emission filters was used to couple excitation into the microscope objective. Fluorescence was collimated using a $f = 200$ mm achromatic doublet (L0, Thorlabs, ACT508-200-A-ML), spectrally split using longpass dichroic mirrors (DM1 and DM2, Chroma Technologies, T570lpxr-UF3 and T635lpxr-UF3), refocused (L1-3, Thorlabs, ACT508-200-A-ML), filtered (Chroma Technologies, ET525/50 m, ET595/44 m and ET706/95 m) and imaged using separate cameras for each wavelength (Andor, Zyla 4.2 Plus) with an effective pixel size of 130 nm (after 2 × 2 binning). The collimation lens and refocusing lenses form a $4f$ configuration for all three light paths. The readout rate was set to 200 MHz. The central $1,024 × 1,024$ pixels ($512 × 512$ after binning) of the camera were used as the region of interest. Alignment between the three channels was performed using an affine transform that was determined using a fluorescent bead sample before each measurement. One microliter TetraSpeck (Invitrogen, T7279) 100-nm-diameter fluorescent beads were added to 300 μl 1 M TRIS buffer, incubated in a custom-built flow chamber[18] for 2 min and washed 2× with 200 μl PBS. Ten regions in this flow chamber were imaged in three color channels simultaneously for 500 frames at 25 ms exposure time, and the fluorescent signal from the individual beads was localized in Picasso Localize, overlaid in Picasso Render and saved with consistent spot identities for all channels. The affine transform determining shift, rotation and skew between channels was determined using a custom script (https://github.com/PhilippSteen/Affine-transformation) based on these spot localizations. The resulting matrices were applied to all collected data thereafter. Cross-talk was quantified using DNA origami with R1 docking sites. In three rounds of Exchange-PAINT, only Cy3B, only Atto643 and only CF488A were imaged on all three cameras (three wavelengths) simultaneously with all three excitation lasers activated. The number of localizations detected in each channel is reported in Supplementary Table 4. The portion of localizations detected in an 'incorrect' channel was below 1% for all cases.

## Excitation power density

The power of the laser beam exiting the objective was determined at the beginning of each measurement day by measuring the intensity in epi-illumination using a Thorlabs S130C power meter. To calculate the power density, the beam profile was recorded in TIRF using a well plate with >20 nM Cy3B imager added. The full FOV ($1,152 × 1,152$ pixels after 2 × 2 binning, $150 × 150$ μm$^2$ area for the single-color microscope and $1,024 × 1,024$ pixels after 2 × 2 binning, and $133 × 133$ μm$^2$ area for three-color microscope) was recorded. A Gaussian was fitted to the beam profile and the average power density across the central FOV used for all other measurements ($576 × 576$ pixels after 2 × 2 binning, $75 × 75$ μm$^2$ area for the single-color microscope and $512 × 512$ pixels after 2 × 2 binning, and $67 × 67$ μm$^2$ area for the three-color microscope) was calculated.

## Image rendering

All microscopy images were rendered in Picasso render[18] using 'individual localization precision, iso' as display setting, which is based on equation 6 from Mortensen et al.[20].

## Quantitative analysis pipeline: DNA origami

**Localize.** Raw TIFF files were loaded into Picasso[18] Localize. The box side length was set to 7 pixels, and the optimal minimum net gradient for spot detection was manually determined for each imaging round. Localizations were calculated using the Gaussian least squares option.

**Render and site identification.** Imaging and alignment files were loaded into Render separately, undrifted by RCC (window size, 1,000 frames), undrifted by picked DNA origami and, finally, resolution permitting, undrifted by picked individual docking sites. The NeNA localization precision was calculated for both imaging and alignment files; the alignment value was used to verify experimental consistency while the imaging round value was reported. Next, the imaging round was aligned to the alignment round. Crosshair DNA origami were picked in the alignment file and exported as picked spots. The picked alignment spot file was loaded into a new render window, and the SMLM clusterer[8] was applied with a minimum number of localizations of 15 and a cluster radius of 0.04 pixels (5.2 nm). The clustered, picked alignment file was then loaded into a custom script that identifies picks with exactly five clusters (that is, correctly folded and complete crosshairs structures) and exports their location. This location file was finally applied to the aligned imaging file, identifying all single docking sites stemming from correctly folded crosshairs DNA origami. These single docking sites were exported for further quantitative analysis.

**Linking localizations.** Analysis was performed using custom code (https://github.com/PhilippSteen/DNA-PAINT_analysis). First, localizations from individual docking sites were linked: successive localizations from one docking site almost certainly stem from one DNA-PAINT binding event. Therefore, each set of one or more successive localizations is considered one binding event.

Given all binding events for a docking site, the mean bright and mean dark times for the docking site were calculated as described previously[15]. Briefly, a cumulative distribution function was created from all dark or bright times for a given docking site, then a single exponential function was fitted to this distribution. Finally, the mean bright or dark time from all single docking sites in a measurement was calculated.

Neither the start nor end of a physical binding event are likely to precisely coincide with the start or end of the respective frames/camera exposure. In other words, an imager may bind while the 100 ms exposure of a frame is half over. Therefore, the first and last frame of a given binding event are not representative of the photon output of the dye in a bound state; they will almost always collect fewer photons than a frame that records a continuously bound imager. Consequently, the photon values reported in this work stem from 'center frames', frames from binding events that are three frames long or longer and are not the first or last frame (Extended Data Fig. 1f). The reported photon values are always photon output per 100 ms.

Finally, by calculating the number of localizations from single docking sites per time window and plotting this over time, the dye stability was estimated.

**SBR calculation.** The reported photon values represent all photons collected over a 7 × 7 pixel area, whereas the background value is a per-pixel offset. The number of photons collected in a one-pixel area, not necessarily corresponding to the physical pixels of the camera, centered at the coordinate of the localization was calculated using the fit parameters that best described the Point Spread Function (PSF). In other words, an area integral over the PSF within the region of one pixel centered at the maximum of the PSF was calculated. This value was then

divided by the background offset, canceling the area term and yielding a unitless SBR. Only 'center frames' as defined above were examined.

**Docking site destruction.** The mean dark time ($\tau_d$) as well as the time elapsed between the final binding event recorded and the end of the measurement ($t_{end}$) were determined for each docking site. The ratio $r = t_{end}/\tau_d$ was calculated for each docking site and all ratios plotted as a histogram.

For a case with no docking site destruction, $r$ is a sampling of an exponential distribution, so the histogram of all $r$ values also follows an exponential distribution (Extended Data Fig. 5).

An exponential decay ($a \times \exp(-(1/\mu) \times r)$) function was fitted to the histogram. For the case of no docking site destruction and no decrease in localization frequency, the mean $\mu$ of the function was equal to 1; the expected time between exponentially distributed binding events and an arbitrary cutoff (end of the measurement) is equal to the mean dark time between binding events. In the case of decreasing numbers of localizations over time, regardless of docking site destruction, the mean of the exponential describing the ratios $r$, $\mu$, was greater than 1.

The percentage of docking sites with ratios $r > 4\mu$ was taken as an approximation of the overall percentage of destroyed docking sites. In a purely exponential distribution with $\mu = 1$, $1/e^4$ (1.83%) of the occurrences lie at $r > 4\mu$. Thus, the reported value was obtained by subtracting 1.83% from the percentage of docking sites with ratios $r > 4\mu$.

### Quantitative analysis pipeline: Nup96
**Localize.** Raw TIFF files were loaded into Picasso Localize. Box side length was set to 9 pixels, and the optimal minimum net gradient for spot detection was manually determined for each imaging round. Localizations were calculated using the Gaussian least squares option. The $Z$ coordinate was fitted using PSF astigmatism.

**Render and NPC selection.** Imaging and alignment files were loaded into Render separately, undrifted by RCC (window size, 1,000 frames) and undrifted by picked individual gold nanoparticles. Next, the imaging round was aligned to the alignment round. Localizations stemming from gold nanoparticles were removed from both files, as their presence would affect the calculated NeNA localization precision, which was subsequently calculated for both files. The alignment round value was used to verify consistent imaging performance, and measurements with poor localization precisions were discarded. More than 50 NPCs were manually picked in the alignment round, then pick similar was used to select a majority of NPCs in the FOV. Misidentified picks were manually removed. The identified NPCs were then exported as picked localizations. Finally, cytoplasmic regions around the nucleus were manually picked and exported.

**Linking localizations.** Localizations from picked NPCs were linked analogously to DNA origami data. 'Center frames' photon values and mean bright times were reported.

**SBR calculation.** The SBR was calculated analogously to DNA origami data.

**Unspecific sticking.** The number of localizations per area from NPCs as well as cytoplasmic picks were calculated for imaging and alignment rounds. The ratio of these, that is, signal/sticking, were calculated for both as well. Finally, the ratio of signal over sticking from the imaging round over signal over sticking from the alignment round was calculated. This yielded a unitless relative specificity that removed bias from imager concentration, off-target bound nanobodies, undesired single-stranded DNA in the nucleus and other effects present for both the dye analyzed and Cy3B in the alignment round. This process is illustrated in Extended Data Fig. 8.

### Quantitative analysis pipeline: Tom20
**Localize.** Raw TIFF files were loaded into Picasso Localize. The box side length was set to 9 pixels, and the optimal minimum net gradient for spot detection was manually determined for each imaging round. Localizations were calculated using the Gaussian least squares option. The $Z$ coordinate was fitted using PSF astigmatism.

**Render and mitochondria selection.** Imaging and alignment files were loaded into Render separately, undrifted by RCC (window size, 1,000 frames) and undrifted by picked individual gold nanoparticles. Next, the imaging round was aligned to the alignment round. Localizations stemming from gold nanoparticles were removed from both files, as their presence would affect the calculated NeNA localization precision, which was subsequently calculated for both files. The alignment round value was used to verify consistent imaging performance, and measurements with poor localization precisions were discarded. The Picasso mask tool was applied to the alignment round with a display pixel size of 100 nm, a blur of 1.0 and a threshold of 0.1. This mask was used to separate mitochondria (saved as one file) and localizations outside of mitochondria (saved as a separate file).

**Unspecific sticking.** The number of localizations from mitochondria was divided by the number of localizations outside of mitochondria for alignment and imaging rounds. Since the areas investigated are identical for alignment and imaging rounds, dividing the 'specificity' from the imaging round by the 'specificity' of the alignment round yields a unitless relative specificity.

### Reporting summary
Further information on research design is available in the Nature Portfolio Reporting Summary linked to this article.

## Data availability
Localization data from this study are available via Zenodo at https://doi.org/10.5281/zenodo.10960858 (ref. 39). Raw microscopy data obtained during this study are available from the corresponding author on reasonable request. Source data are provided with this paper.

## Code availability
Raw image processing was performed using Picasso, available via GitHub at https://github.com/jungmannlab/picasso with documentation provided at https://picassosr.readthedocs.io/en/latest/render.html. Custom software is available via Zenodo at https://doi.org/10.5281/zenodo.10960858 (ref. 39) as well as via GitHub at https://github.com/PhilippSteen/DNA-PAINT_analysis and https://github.com/PhilippSteen/Affine-transformation.

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

## Acknowledgements

This research was funded in part by the European Research Council through an ERC Consolidator Grant (ReceptorPAINT, Grant agreement number 101003275), the BMBF (Project IMAGINE, FKZ: 13N15990), the Volkswagen Foundation through the initiative 'Life?—A Fresh Scientific Approach to the Basic Principles of Life' (grant no. 98198), the Max Planck Foundation and the Max Planck Society. We thank the Ries and Ellenberg groups for the kind gift of the Nup96-GFP cell line; the Schnermann and Tinnefeld labs for the kind gift of Cy5B; and S. Strauss and E. Falgenhauer for vital assistance conjugating labeling probes and dyes. We thank M. K. Steen-Mueller and U. Mueller for proofreading the paper. We thank V. Glembockyte, M. Scheckenbach and J.-H. Krohn for fruitful discussions. P.R.S. and E.M.U. acknowledge support by the IMPRS-ML graduate school. L.A.M. acknowledges a postdoctoral fellowship from the European Union's Horizon 20212022 research and innovation program under Marie Skłodowska-Curie grant agreement no. 101065980. F.O. was supported by Deutsche Forschungsgemeinschaft (DFG) through the SFB1286 (project Z04).

## Author contributions

P.R.S., E.M.U., L.A.M. and R.J. designed experiments and analysis methods. P.R.S. and E.M.U. performed experiments. P.R.S., E.M.U., J.K., A.P., K.J. and E.F.F. prepared samples. F.O. provided labeling reagents. P.R.S. and L.A.M. designed and constructed optical systems. P.R.S. analyzed the data. P.R.S. and R.J. wrote the paper with input from L.A.M. and E.M.U. R.J. conceived and supervised the study. All authors reviewed and approved the paper.

## Funding

## Competing interests

F.O. is a shareholder of NanoTag Biotechnologies GmbH. The other authors declare no competing interests.

## Additional information

**Extended data** is available for this paper at https://doi.org/10.1038/s41592-024-02374-8.

**Correspondence and requests for materials** should be addressed to Ralf Jungmann.

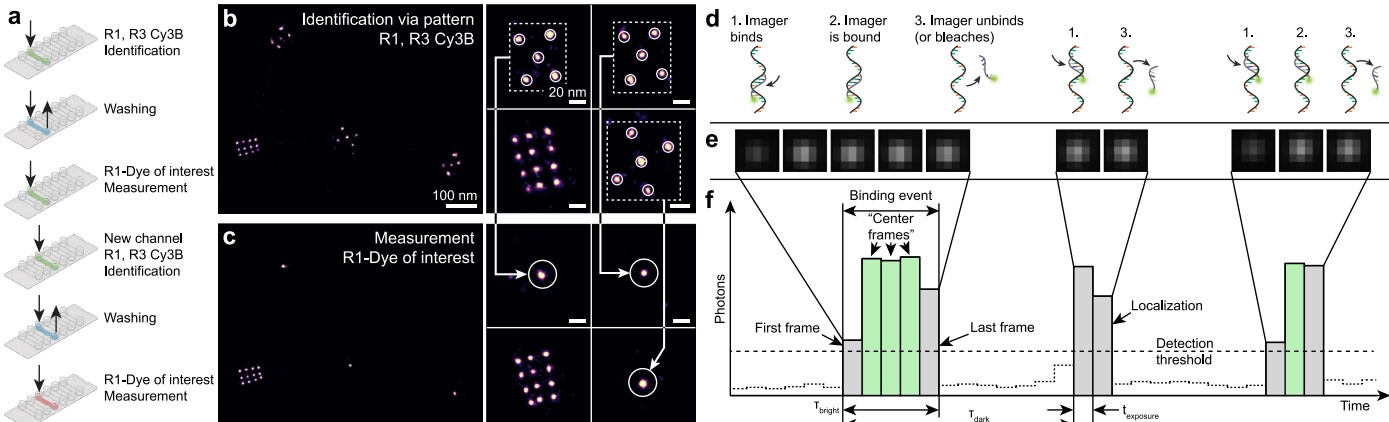

**Extended Data Fig. 1 | DNA origami data acquisition and analysis. a,** Each dye is imaged in a separate flow chamber. First, R1 and R3 Cy3B imagers are used to acquire docking sites of interest and barcode sites. Next, all imagers are removed and only the sites of interest are imaged with the dye of interest. The process is repeated for a different dye in a different chamber. **b,** Localizations stemming from R1 (sites of interest) and R3 (barcodes) are used to identify DNA origami featuring individual docking sites. **c,** The dye of interest imaging round is aligned to the identification round using 20 nm DNA origami. The single docking sites are used for quantitative analysis. **d,** Individual docking sites experience DNA-PAINT imager binding and unbinding. **e,** Such bound states are recorded by the sCMOS camera and localized. **f,** Plotted as an intensity-time trace, localizations at the beginning and end of a binding event frequently exhibit lower photon counts than others. This is due to the imagers binding or unbinding during a single camera exposure time, leading to fewer collected photons. For downstream photon analysis, only 'center frames' are used as they are representative of the dye's photon output.

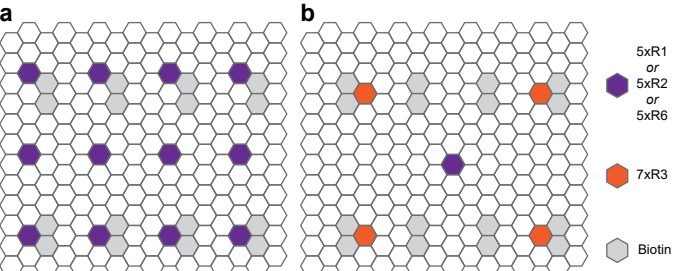

**Extended Data Fig. 2 | DNA origami designs. a**, Twelve 20 nm-spaced docking sites used to illustrate achievable resolution for R1, R2 or R6 imager strands. **b**, Single docking site for R1, R2 or R6 imagers with four 7xR3 barcoding / identification sites for Exchange-PAINT-based identification of the single docking sites.

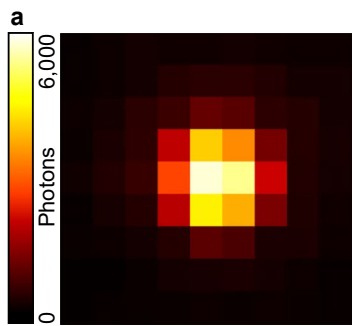

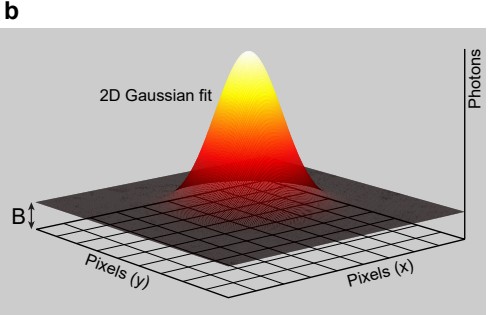

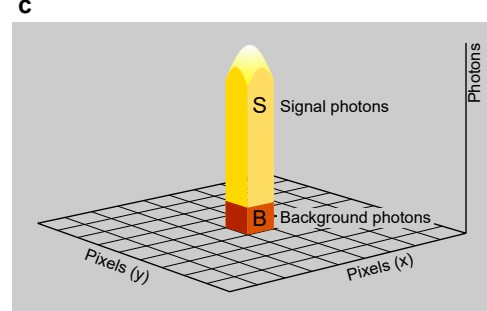

**Extended Data Fig. 3 | Photon count and SBR calculation. a**, Photons from a single dye as detected by a sCMOS camera in a single frame (100 ms exposure). **b**, A 2D Gaussian fit is applied to the photon distribution. The volume under the Gaussian, without the background offset B, is recorded as the number of photons for this given localization. **c**, The signal to background ratio (SBR) is calculated by dividing the number of non-background photons S in the pixel centered in the maximum of the Gaussian fit by the background photon value B.

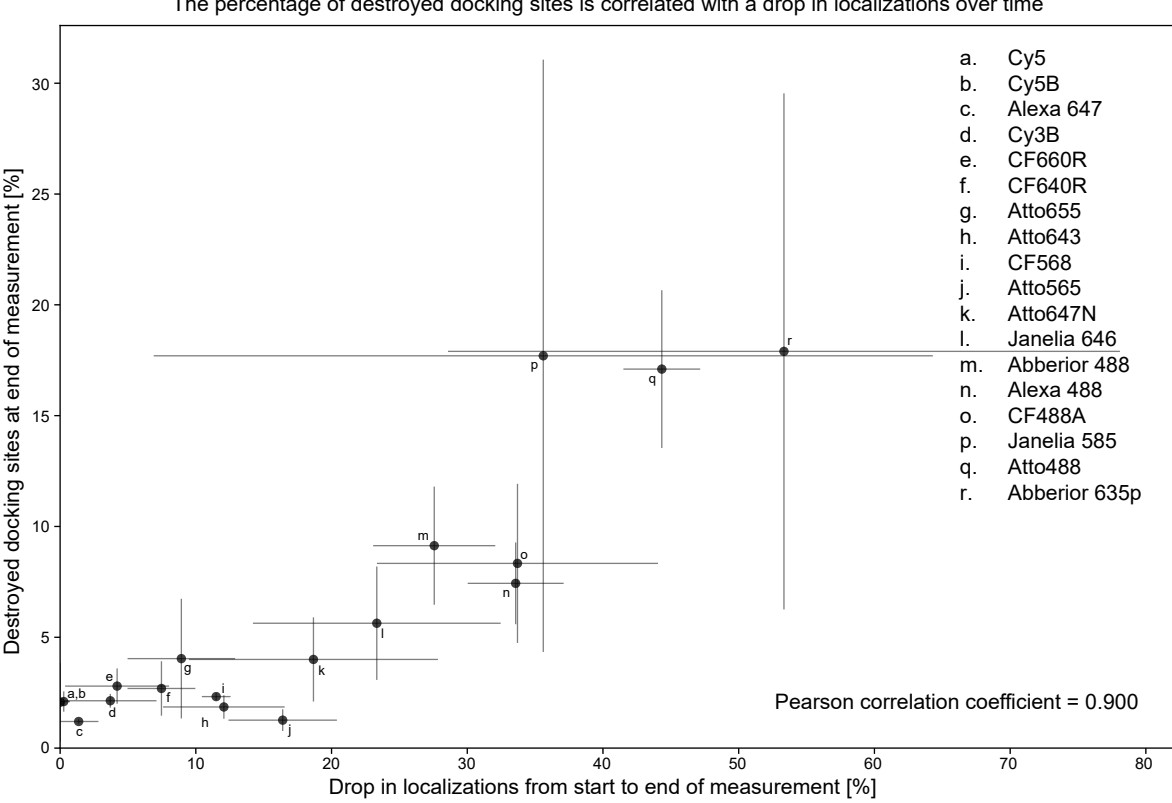

**Extended Data Fig. 4 | Docking site destruction vs. decrease in number of localizations.** The percentage of docking sites destroyed by the end of the measurement is plotted against the drop in localizations over the course of the measurement. Dots represent the mean, and error bars represent s.d. across N=3 repeats.

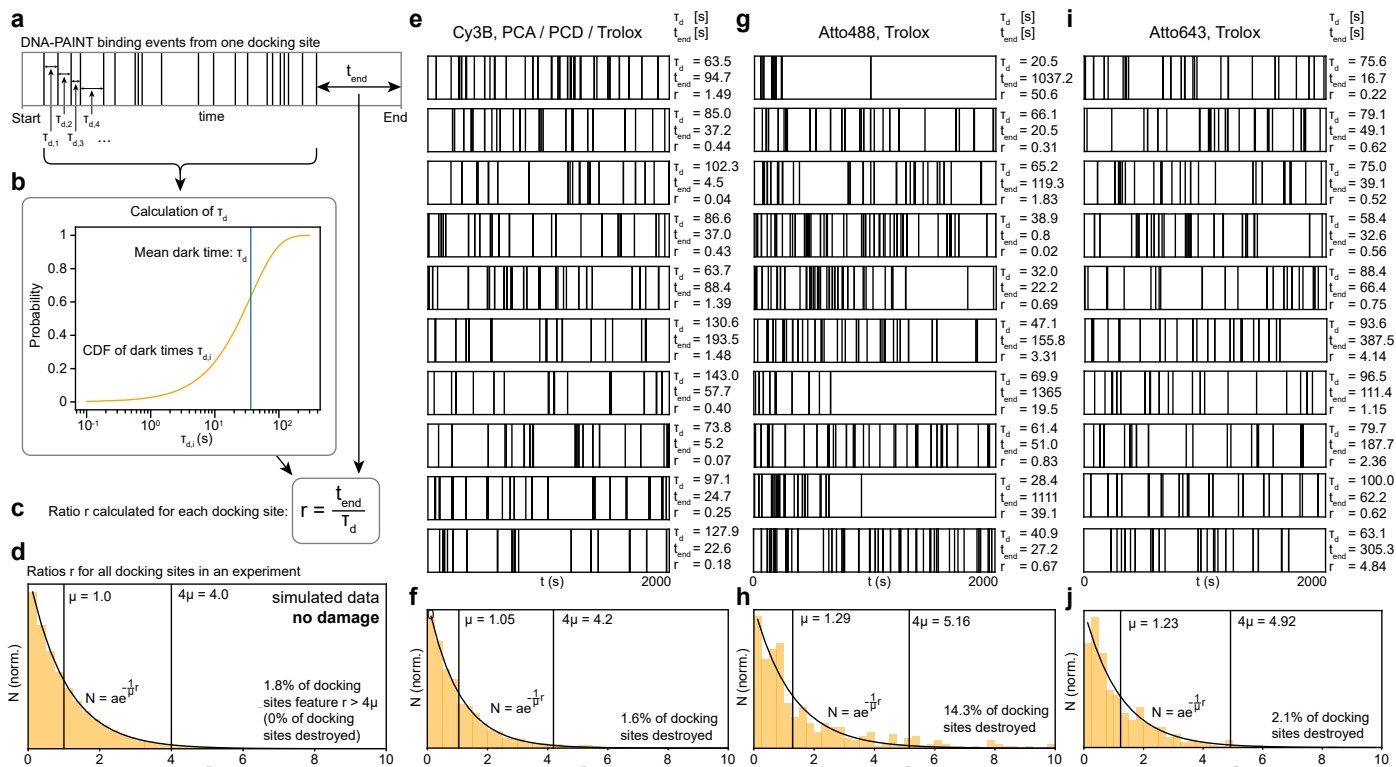

**Extended Data Fig. 5 | Estimation of docking site destruction. a**, DNA-PAINT features repeated binding events of imagers to one docking site. Here, the time between binding events is defined as $\tau_{d,i}$ and the time between the final recorded binding event and the end of the measurement is defined as $t_{end}$. **b**, The mean dark time $\tau_d$ for one docking site is calculated using the CDF of the exponentially distributed times between binding events $\tau_{d,i}$. **c**, r is defined as the ratio of the time between the final binding event and the end of the measurement over the mean dark time. **d**, Plotting the ratios for all docking sites in a measurement as a histogram yields an exponential decay that is described by the function $N = ae^{-\frac{1}{\mu}r}$. Simulated data with no damage or destruction to docking sites results in $\mu=1$. **e**, Typical DNA-PAINT binding events for Cy3B on DNA origami with PCA, PCD and Trolox. The mean dark time, $t_{end}$ and r are reported for each trace.

**f**, The histogram of the r values follows an exponential decay. The percentage of docking sites with r values greater than $4\mu$, minus the portion of the exponential distribution extending beyond $4\mu$, are reported as an approximation of docking site destruction. **g**, Atto488 measured with Trolox exhibits evidently damaged or destroyed docking sites. **h**, The histogram representation yields a higher percentage of destroyed docking sites than Cy3B, namely 14.3%. **i**, Atto643 with Trolox features fewer destroyed docking sites. **j**, This is confirmed by the histogram of r values, however the mean $\mu$ of the distribution being greater than 1 is indicative of a reduction of sampling over the course of the measurement, which in turn is corroborated by the reduction of localizations over the course of the measurement (see Table 1, 'Localization Drop').

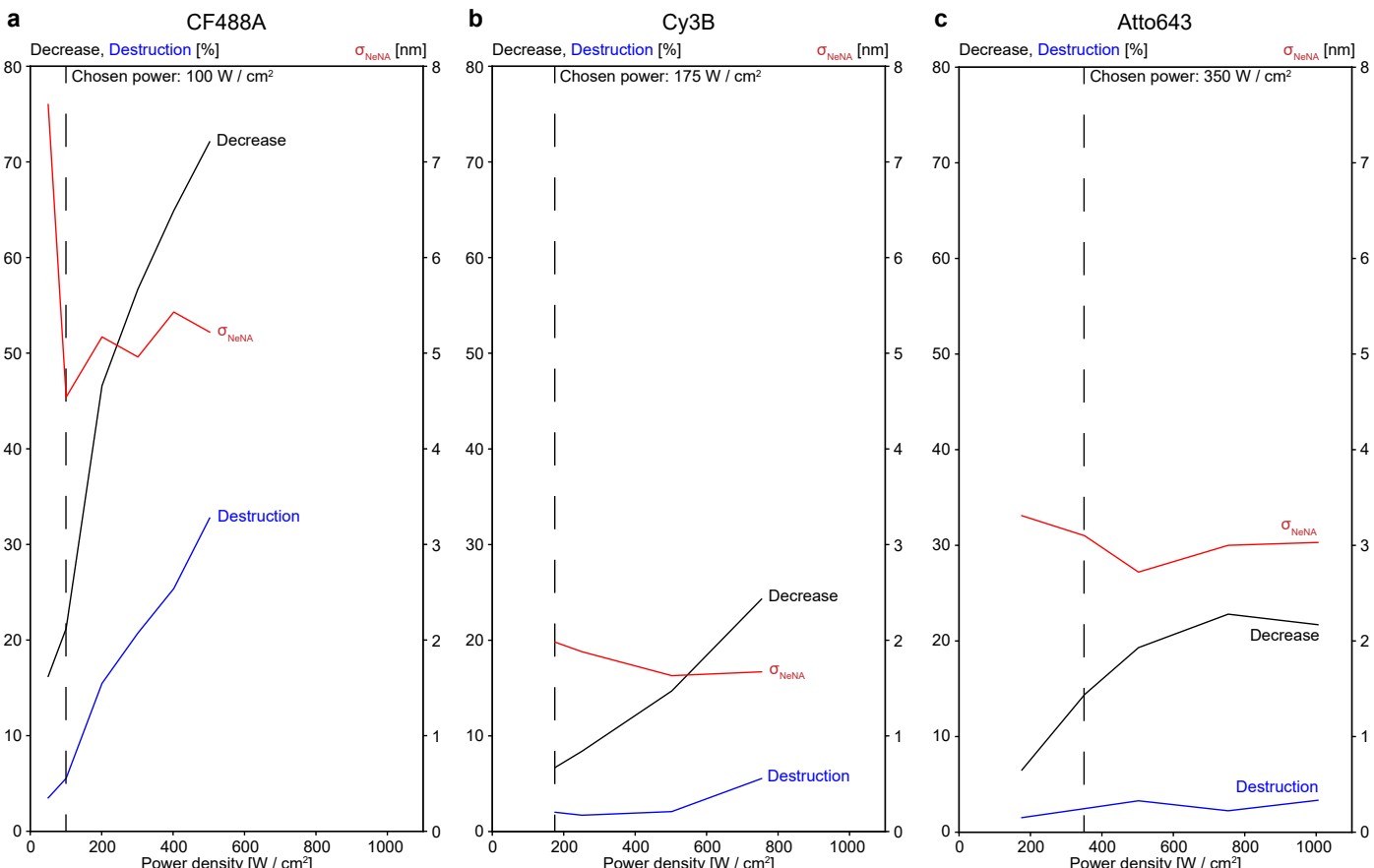

**Extended Data Fig. 6 | Dye performance as a function of excitation power density.** The decrease in localizations over time (black), percentage of destroyed binding sites (blue) and localization precision (red) of CF488A (**a**), Cy3B (**b**) and Atto643 (**c**) for docking sites on DNA origami are plotted against excitation power density. The power density chosen for all other DNA origami benchmarking measurements is indicated by the vertical dashed line. This power density was chosen as a compromise of decrease / damage and good localization precision.

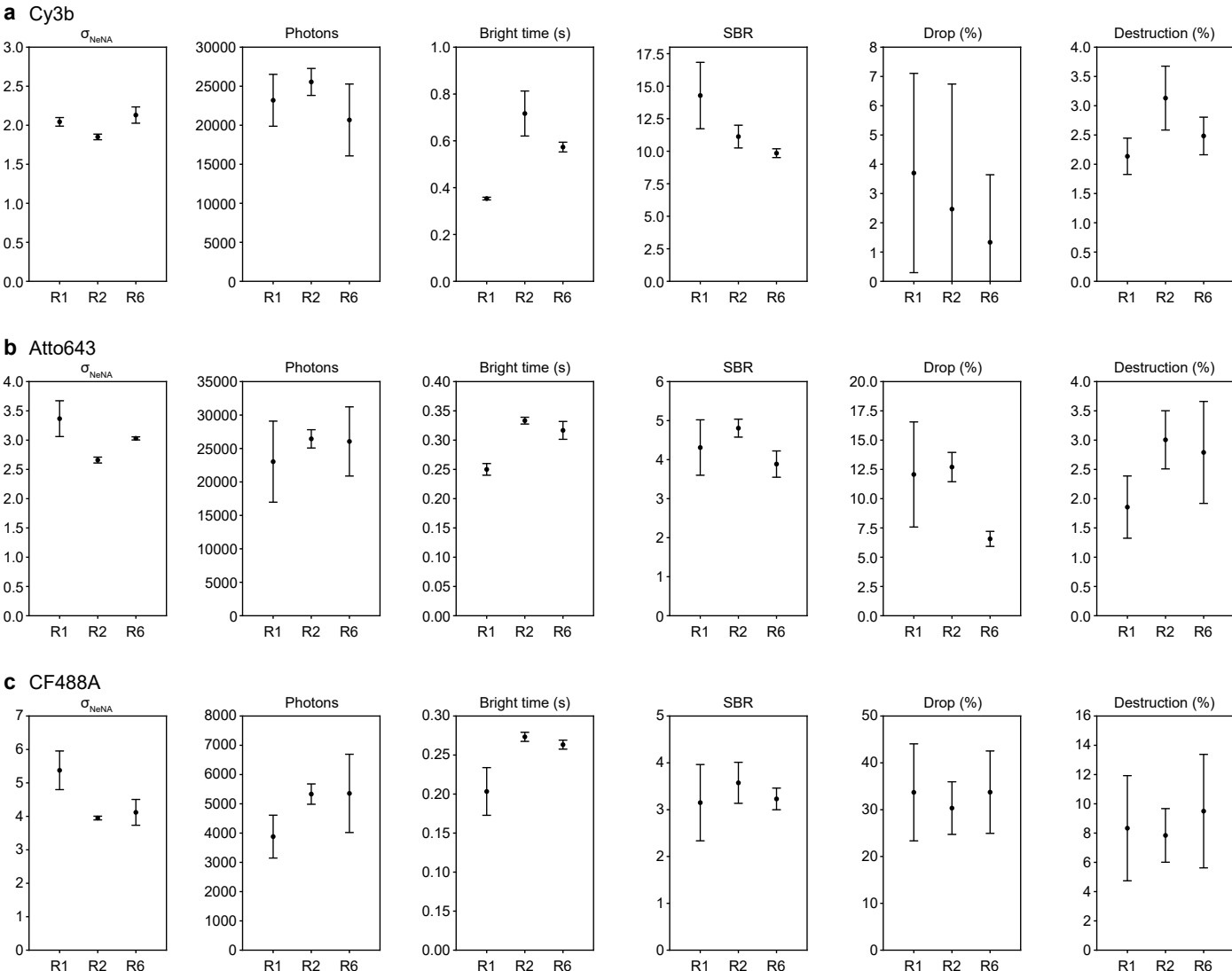

**Extended Data Fig. 7 | Sequence comparison. a**, Resolution, photon count, bright time, signal to background ratio (SBR), drop of localizations over the course of the measurement and percentage of destroyed docking sites at the end of the measurement for three Cy3B-labeled DNA-PAINT imager sequences (R1, R2, R6). **b**, As above, using Atto643. **c**, As above, using CF488A. Error bars represent s.d. across N=3 repeats.

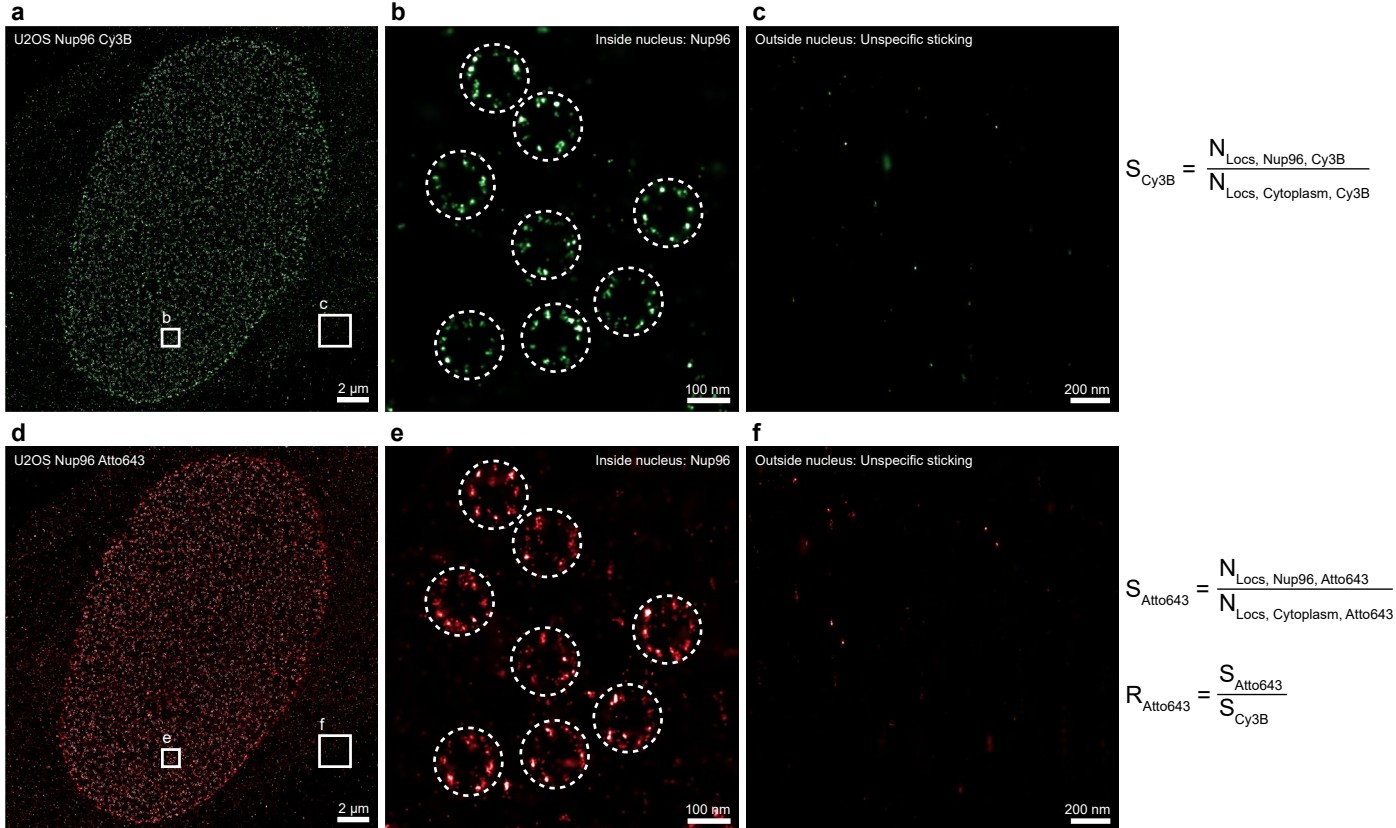

**Extended Data Fig. 8 | Relative Dye Specificity. a**, Fixed U2OS-Nup96-mEGFP cells were labeled with R1 anti-GFP nanobodies and imaged using Cy3B. **b**, NPCs can be clearly identified and picked. Localizations originating from NPCs are defined as 'signal'. The number of 'signal' localizations per area is $N_{Locs, Nup96, Cy3B}$. **c**, Far fewer localizations occur in the cytoplasm, outside the labeled region. These localizations are defined as 'sticking'. The number of 'sticking' localizations per area is $N_{Locs, Cytoplasm, Cy3B}$. The ratio of $N_{Locs, Nup96, Cy3B}$ over $N_{Locs, Cytoplasm, Cy3B}$ yields the specificity $S_{Cy3B}$ that Cy3B achieved in this measurement. **d**, The same cell was imaged using a different dye, in this case Atto643. **e**, The exact same NPCs as selected in imaging round 1 (**a-c**) are used in the Atto643 round, localizations originating from these picks are defined as 'signal' as above. The number of 'signal' localizations per area is $N_{Locs, Nup96, Atto643}$. **f**, The exact same cytoplasmic regions as in imaging round 1 are selected, these localizations are defined as 'sticking'. The number of 'sticking' localizations per area is $N_{Locs, Cytoplasm, Atto643}$. The ratio of $N_{Locs, Nup96, Atto643}$ over $N_{Locs, Cytoplasm, Atto643}$ yields the specificity $S_{Atto643}$ that Atto643 achieved in this measurement. The ratio of specificity $S_{Atto643}$ over $S_{Cy3B}$ is defined as the relative specificity $R_{Atto643}$. This process was applied to N=3 cells per dye.

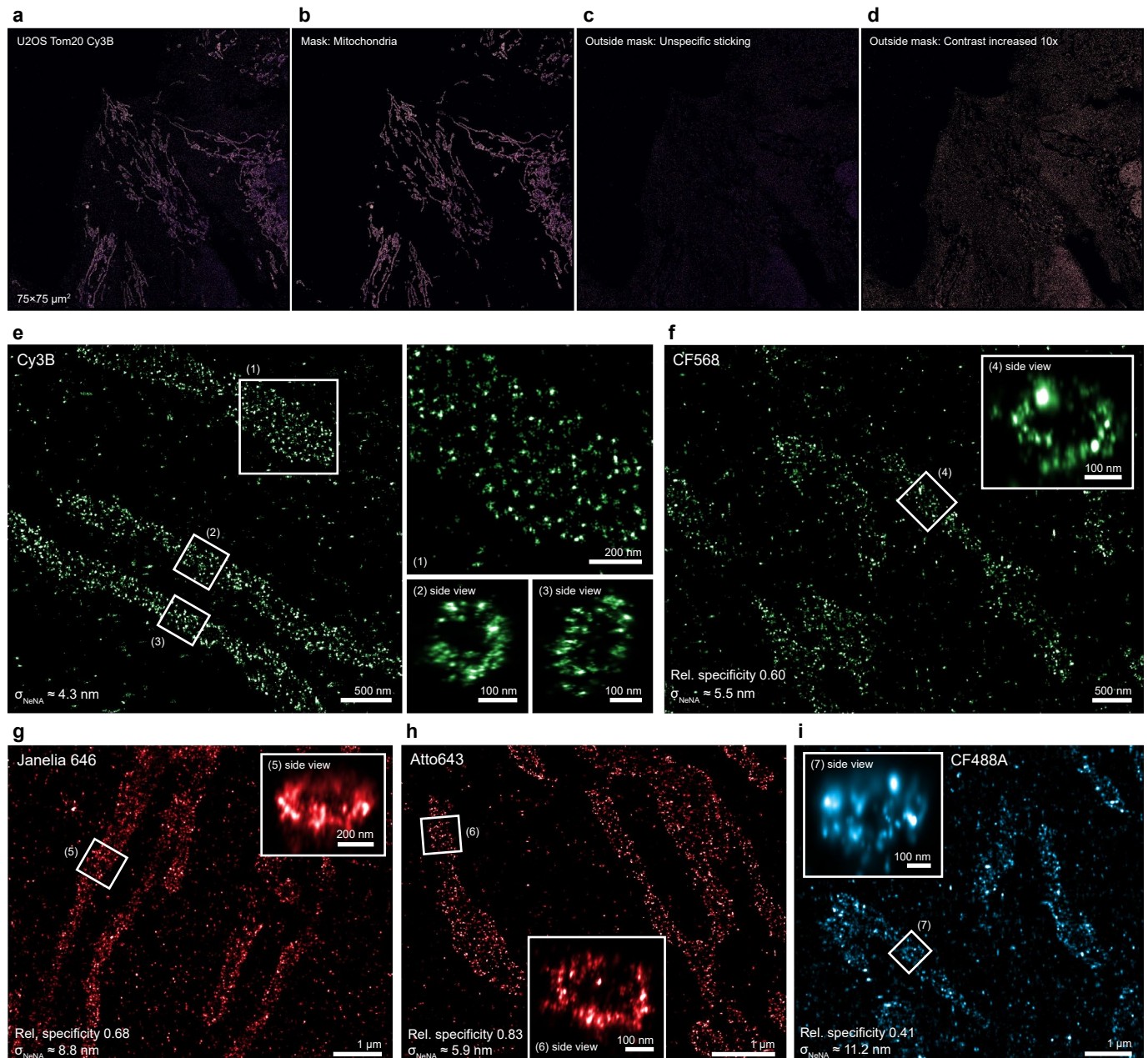

**Extended Data Fig. 9 | Cellular imaging with Tom20. a**, Fixed U2OS cells were labeled with Anti-Tom20 rabbit antibodies and R1 DNA-labeled anti-rabbit nanobodies, labeling mitochondria, and imaged with Cy3B. **b**, Mitochondria were identified using the Picasso Render mask tool. **c**, The area outside the mask features unspecific imager sticking. **d**, Since there is far less off-target sticking than on-target localizations, a representation with a 10x increase in contrast is shown. Dye specificity in a given measurement is estimated by dividing the number of localizations in mitochondria over the number of localizations outside. **e**, Cy3B achieves single-label resolutions (insert 1) and clearly resolves the hollow structure in 3D (inserts 2 and 3). **f**, CF568 achieves a slightly lower resolution and also features more off-target localizations as compared to Cy3B (relative specificity of 0.6), consistent with the results obtained from imaging the NPC. **g**, Janelia 646, **h**, Atto643 as well as **i**, CF488A all feature comparable trends in resolution and specificity when imaging Tom20 as compared to NPC imaging. Imaging was repeated 3 times with similar results, one dataset is shown.

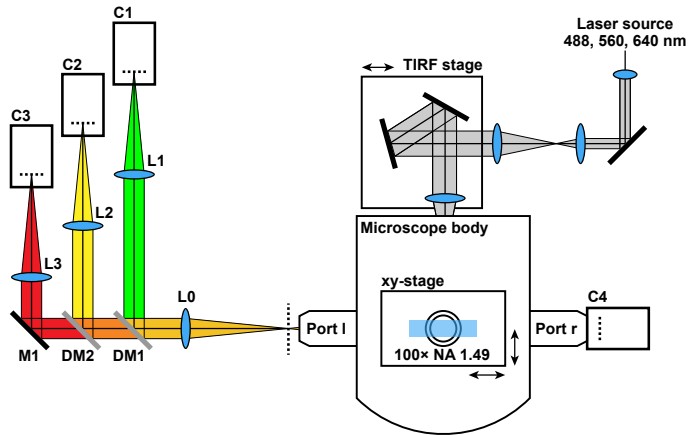

**Extended Data Fig. 10 | Three-color microscope setup.** L0, L1, L2, L3: f=200 mm achromatic doublet (Thorlabs, ACT508-200-A-ML). DM1: Chroma Technologies, T570lpxr-UF3. DM2: Chroma Technologies, T635lpxr-UF3. C1, C2, C3 Andor Zyla 4.2 plus.

# Reporting Summary

## Statistics

For all statistical analyses, confirm that the following items are present in the figure legend, table legend, main text, or Methods section.

| n/a | Confirmed | |
|---|---|---|
| ☐ | ☒ | The exact sample size (*n*) for each experimental group/condition, given as a discrete number and unit of measurement |
| ☐ | ☒ | A statement on whether measurements were taken from distinct samples or whether the same sample was measured repeatedly |
| ☒ | ☐ | The statistical test(s) used AND whether they are one- or two-sided *Only common tests should be described solely by name; describe more complex techniques in the Methods section.* |
| ☒ | ☐ | A description of all covariates tested |
| ☒ | ☐ | A description of any assumptions or corrections, such as tests of normality and adjustment for multiple comparisons |
| ☐ | ☒ | A full description of the statistical parameters including central tendency (e.g. means) or other basic estimates (e.g. regression coefficient) AND variation (e.g. standard deviation) or associated estimates of uncertainty (e.g. confidence intervals) |
| ☒ | ☐ | For null hypothesis testing, the test statistic (e.g. *F*, *t*, *r*) with confidence intervals, effect sizes, degrees of freedom and *P* value noted *Give P values as exact values whenever suitable.* |
| ☒ | ☐ | For Bayesian analysis, information on the choice of priors and Markov chain Monte Carlo settings |
| ☒ | ☐ | For hierarchical and complex designs, identification of the appropriate level for tests and full reporting of outcomes |
| ☒ | ☐ | Estimates of effect sizes (e.g. Cohen's *d*, Pearson's *r*), indicating how they were calculated |

*Our web collection on statistics for biologists contains articles on many of the points above.*

## Software and code

Policy information about availability of computer code

Data collection: Raw microscopy data was acquired using μManager version 2.0.1 (Edelstein, A., Amodaj, N., Hoover, K., Vale, R. & Stuurman, N. Curr. Protoc. Mol. Biol. 14.20 (2010)).

Data analysis: All data were analyzed using the open source software Picasso versions 0.6.0 - 0.6.9 (Schnitzbauer, J., Strauss, M. T., Schlichthaerle, T., Schueder, F., & Jungmann, R. (2017). Super-resolution microscopy with DNA-PAINT. Nature Protocols, 12(6), 1198–1228. http://doi.org/10.1038/nprot.2017.024), and custom code (10.5281/zenodo.10960858 or https://github.com/PhilippSteen/DNA-PAINT_analysis and https://github.com/PhilippSteen/Affine-transformation)

For manuscripts utilizing custom algorithms or software that are central to the research but not yet described in published literature, software must be made available to editors and reviewers. We strongly encourage code deposition in a community repository (e.g. GitHub). See the Nature Portfolio guidelines for submitting code & software for further information.

## Data

Policy information about availability of data

All manuscripts must include a data availability statement. This statement should provide the following information, where applicable:
- Accession codes, unique identifiers, or web links for publicly available datasets
- A description of any restrictions on data availability
- For clinical datasets or third party data, please ensure that the statement adheres to our policy

> Localization data from this study are available at 10.5281/zenodo.10960858. Raw microscopy data obtained during this study are available from the corresponding author on reasonable request.

## Human research participants

Policy information about studies involving human research participants and Sex and Gender in Research.

| | |
|---|---|
| Reporting on sex and gender | n/a |
| Population characteristics | n/a |
| Recruitment | n/a |
| Ethics oversight | n/a |

Note that full information on the approval of the study protocol must also be provided in the manuscript.

# Field-specific reporting

Please select the one below that is the best fit for your research. If you are not sure, read the appropriate sections before making your selection.

☒ Life sciences  ☐ Behavioural & social sciences  ☐ Ecological, evolutionary & environmental sciences

For a reference copy of the document with all sections, see nature.com/documents/nr-reporting-summary-flat.pdf

# Life sciences study design

All studies must disclose on these points even when the disclosure is negative.

| | |
|---|---|
| Sample size | Sample size n is defined as the number of independent microscopy experiments performed. |
| Data exclusions | No data were excluded. |
| Replication | All replications (3 independent experiments per parameter investigated) were successful. |
| Randomization | n/a, no grouping of experiments or samples was performed. |
| Blinding | n/a, no grouping of experiments or samples was performed. |

# Reporting for specific materials, systems and methods

We require information from authors about some types of materials, experimental systems and methods used in many studies. Here, indicate whether each material, system or method listed is relevant to your study. If you are not sure if a list item applies to your research, read the appropriate section before selecting a response.

## Materials & experimental systems

| n/a | Involved in the study |
|-----|----------------------|
| ☐ | ☒ Antibodies |
| ☐ | ☒ Eukaryotic cell lines |
| ☒ | ☐ Palaeontology and archaeology |
| ☐ | ☒ Animals and other organisms |
| ☒ | ☐ Clinical data |
| ☒ | ☐ Dual use research of concern |

## Methods

| n/a | Involved in the study |
|-----|----------------------|
| ☒ | ☐ ChIP-seq |
| ☒ | ☐ Flow cytometry |
| ☒ | ☐ MRI-based neuroimaging |

## Antibodies

**Antibodies used**

1) Mouse monoclonal (SAP7F407) anti-Bassoon, Enzo (Cat# ADI-VAM-PS003-F; RRID:AB_11181058), dilution 1 in 200
2) Rabbit polyclonal anti-VGAT, Invitrogen (Cat# PA5-27569; RRID:AB_2545045), dilution 1 in 300
3) Rabbit monoclonal (EPR15581-54) anti-Tom20, Abcam (Cat# ab186735; RRID:AB_2889972), dilution 1 in 200
4) Mouse monoclonal (69H10) anti-Neurofilament L, Synaptic Systems (Cat# 171011; RRID:AB_2891275), dilution 1 in 200
5) Mouse monoclonal (42/B) anti-βII Spectrin, BD Biosciences (Cat# 612562; RRID:AB_399853), dilution 1 in 100
6) sdAb anti-PSD95 (1B2), NanoTag Biotechnologies (Cat# N3705), dilution 1 in 200
7) sdAb anti-GFP (1H1), NanoTag Biotechnologies (Cat# N0305), dilution 1 in 200
8) sdAb anti-Mouse IgG (10A4), NanoTag Biotechnologies (Cat# N2005), dilution 1 in 300
9) sdAb anti-Rabbit IgG (10E10), NanoTag Biotechnologies (Cat# N2405), dilution 1 in 300
10) Multiplexing Blocker Mouse, NanoTag Biotechnologies (Cat# K0102-50), dilution 1 in 200
11) Multiplexing Blocker Mouse, NanoTag Biotechnologies (Cat# K0202-50), dilution 1 in 200

**Validation**

Antibodies and Nanobodies were verified by Manufacturers by Immunofluorescence and Western blot to ensure that the binders bind to the antigen stated.

## Eukaryotic cell lines

Policy information about cell lines and Sex and Gender in Research

**Cell line source(s)**

U-2 OS-CRISPR-Nup96-mEGFP cells were obtained from the Ellenberg and Ries lab (Reference: https://doi.org/10.1038/s41592-019-0574-9).

**Authentication**

The cell lines were not authenticated.

**Mycoplasma contamination**

All cell lines have been tested negative for mycoplasma contamination.

**Commonly misidentified lines**
(See ICLAC register)

No commonly misidentified cell lines were used.

## Animals and other research organisms

Policy information about studies involving animals; ARRIVE guidelines recommended for reporting animal research, and Sex and Gender in Research

**Laboratory animals**

Wild-type Wistar rat pregnant mothers and pups (Rattus norvegicus), P2.

**Wild animals**

No wild animals were used in this study.

**Reporting on sex**

Animals of both sexes were used in this study.

**Field-collected samples**

No field-collected samples were used in this study.

**Ethics oversight**

Animal experiments were approved by the local authority, the Lower Saxony State Office for Consumer Protection and Food Safety (Niedersächsisches Landesamt für Verbraucherschutz und Lebensmittelsicherheit).

Note that full information on the approval of the study protocol must also be provided in the manuscript.

