## [Peer Review File · Nature Methods]

Peer Review Information

Manuscript Title: The DNA-PAINT palette: A comprehensive performance analysis of fluorescent dyes

Corresponding author name(s): Ralf Jungmann

Editorial Notes: None

Reviewer Comments & Decisions:

Decision Letter, initial version:

Dear Ralf,

Your Analysis, "The DNA-PAINT palette: A comprehensive performance analysis of fluorescent dyes", has now been seen by three reviewers. As you will see from their comments below, although the reviewers find your work of considerable potential interest, they have raised a number of concerns. We are interested in the possibility of publishing your paper in Nature Methods, but would like to consider your response to these concerns before we reach a final decision on publication.

We therefore invite you to revise your manuscript to address these concerns. We think the reviews are generally constructive and addressing them should improve the paper. In terms of additional analysis, the referees want your team to assess whether the DNA sequence/dye combo matters, whether dye performance is the same across targets, and what impact exposure time/illumination intensity has on dye performance. We think these should be addressable in a reasonable timescale, but please let us know if any of them are unnecessarily challenging or unlikely to yield a meaningful result. We also ask that you clarify some of the confusion shared by the refs regarding how some of your assays/calculations were done and have a slightly more rigorous treatment of background labeling.

We are committed to providing a fair and constructive peer-review process. Again, please do not hesitate to contact us if there are specific requests from the reviewers that you believe are technically impossible or unlikely to yield a meaningful outcome.

* include a point-by-point response to the reviewers and to any editorial suggestions

* please underline/highlight any additions to the text or areas with other significant changes to facilitate review of the revised manuscript

- * address the points listed described below to conform to our open science requirements
- * ensure it complies with our general format requirements as set out in our guide to authors at www.nature.com/naturemethods
- * resubmit all the necessary files electronically by using the link below to access your home page

[Redacted]

We hope to receive your revised paper within three months. If you cannot send it within this time, please let us know. In this event, we will still be happy to reconsider your paper at a later date so long as nothing similar has been accepted for publication at Nature Methods or published elsewhere.

OPEN SCIENCE REQUIREMENTS

REPORTING SUMMARY AND EDITORIAL POLICY CHECKLISTS

IMAGE INTEGRITY

When submitting the revised version of your manuscript, please pay close attention to our Digital Image Integrity Guidelines and to the following points below:

DATA AVAILABILITY

We strongly encourage you to deposit all new data associated with the paper in a persistent repository where they can be freely and enduringly accessed. We recommend submitting the data to discipline-specific and community-recognized repositories; a list of repositories is provided here:

<http://www.nature.com/sdata/policies/repositories>

All novel DNA and RNA sequencing data, protein sequences, genetic polymorphisms, linked genotype and phenotype data, gene expression data, macromolecular structures, and proteomics data must be deposited in a publicly accessible database, and accession codes and associated hyperlinks must be provided in the "Data Availability" section.

Please include a "Data availability" subsection in the Online Methods. This section should inform readers about the availability of the data used to support the conclusions of your study, including accession codes to public repositories, references to source data that may be published alongside the paper, unique identifiers such as URLs to data repository entries, or data set DOIs, and any other statement about data availability. At a minimum, you should include the following statement: "The data that support the findings of this study are available from the corresponding author upon request", describing which data is available upon request and mentioning any restrictions on availability. If DOIs are provided, please include these in the Reference list (authors, title, publisher (repository name), identifier, year). For more guidance on how to write this section please see:

<http://www.nature.com/authors/policies/data/data-availability-statements-data-citations.pdf>

CODE AVAILABILITY

Please include a "Code Availability" subsection in the Online Methods which details how your custom code is made available. Only in rare cases (where code is not central to the main conclusions of the paper) is the statement "available upon request" allowed (and reasons should be specified).

For more information on our code sharing policy and requirements, please see:
<https://www.nature.com/nature-research/editorial-policies/reporting-standards#availability-of-computer-code>

MATERIALS AVAILABILITY

ORCID

Nature Methods is committed to improving transparency in authorship. As part of our efforts in this direction, we are now requesting that all authors identified as 'corresponding author' on published papers create and link their Open Researcher and Contributor Identifier (ORCID) with their account on the Manuscript Tracking System (MTS), prior to acceptance. This applies to primary research papers only. ORCID helps the scientific community achieve unambiguous attribution of all scholarly contributions. You can create and link your ORCID from the home page of the MTS by clicking on 'Modify my Springer Nature account'. For more information please visit please visit www.springernature.com/orcid.

Sincerely,
Rita

Rita Strack, Ph.D.
Senior Editor

Nature Methods

Reviewers' Comments:

Reviewer #1:

Remarks to the Author:

In this manuscript the authors characterize a library of 18 different fluorescent dyes for use in DNA-PAINT experiments. The dyes are first analyzed on a well-behaved DNA origami system to obtain their fluorescent output and signal-background ratio (and the consequent spatial resolution), as well as the dyes' impact on site stability. All dyes were then tested in a cellular experiment imaging a Nup96 nuclear pore complex to obtain how these properties would persist in a more complicated environment, and to additionally quantify the dyes' propensity for (undesired) non-specific binding. Lastly, three of the best-performing dyes corresponding to the different excitation channels were selected for use in a multiplexing experiment to image neurons, verifying that high-resolution imaging could be performed with simultaneous measurements involving dye-mixtures. While the methodologies are now routine in the field (in fact the DNA paint calibration markers represent one of the first commercial applications of DNA origami structures), the fundamental science is sound, the presentation of the data is clear, and, most importantly, the utility of such a report as a resource for future DNA-PAINT experiments is certainly beneficial to the single-molecule and broader scientific community as a whole. I believe there are a few areas the current paper could be improved in order to become more accessible to the broader scientific community and to reinforce the results displayed in the figures which are parsed into "major" and "minor" critiques below. As Nature Methods is a journal which publishes "performance comparisons of related, established methods", this paper falls well within the boundaries of the journal, and I would recommend publication after addressing the critiques outlined below.

Major Critique 1: Quantification and discussion of dye-dependent docking site destruction

This critique comes in two parts: 1) the visual clarity of how docking-site stability was calculated and 2) a discussion of how important the docking-site stability parameter is compared to other dye-related parameters within DNA PAINT experiments.

First (1), it is difficult to visualize how the docking-site stability reported (as in Figure 2 e-g, Table 1, etc.) is calculated. The methods section states that "The ratio $r = \tau_{end}/\tau_d$ was calculated for each docking site and all ratios plotted as a histogram.", however these histograms (and their corresponding exponential fits) are not shown, and neither are the relevant μ values and their related cutoffs. Since the authors have access to this data I believe it should be reported as sub-figures in the Supporting Information, which would aid the reader in visualizing how well the r-histograms fit the exponential decay, how strict the 4μ cutoff is, etc. Currently the reader only has access to the "final result" as reported in Table 1. Additionally, the y-axes describing "locs per 20 s per d. site" (in Fig 2 e-g and all SI figures) is not described, and the relation between "Localization drop" and "Destroyed binding sites" is not defined or explored. It is not intuitive to me that some dyes would result in binding sites being destroyed (Cy5, Cy3b, etc.) while showing fewer (or no) "Localization drop" (would binding-site destruction not cause localizations to drop?). I assume that this discrepancy is due simply to statistical variance, yet the differences between the "Destroyed sites" and "Localization drop" values at the end of Table 1 are, in some cases, well outside the variance reported, and as the intermediate data is not presented in any other places it is difficult to see why that is the case. I would like to see the intermediate data (r-histograms, fits, μ values, etc.) presented somewhere, and for the

authors to include a short discussion in either the methods section or the SI describing these parameters, as they are relevant to the binding-site stability portion of the paper.

Second (2) it seems to me a large percentage of the target-readers for this paper would be labs that are “beginners” at DNA-PAINT and are selecting potential dyes for use in their future experiments. In other words, the readers are labs which might not know how significant a 4.0% loss of binding sites is over 2,000 seconds compared to a precision difference of 2 nm at 70 mW excitation power. I believe the accessibility of this paper would be significantly improved by a brief discussion about the relative importance of these various dye-dependent parameters, in addition to the (already included) selection by the authors of the “best-performing dyes”.

Major Critique 2: Reporting of the “relative dye specificity in an individual measurement”

It was not 100% clear to me exactly how the “dye specificity in an individual measurement” was calculated, so allow me to describe what I think the authors did to calculate this parameter: For each “dye of interest” the authors ran TWO sequential DNA-PAINT experiments (runs), FIRST a run with Cy3B as a reference, then a wash, then a SECOND run with the “dye of interest”. From the nuclear images of the FIRST run (as represented top of Figure 3g-i), the authors manually selected several ROIs describing Nup96s before algorithmically finding all such ROIs, they then manually reviewed these ROIs to ensure accuracy. The authors also moved the FOV to image the cell away from the nucleus which should contain no Nup96, but might contain some unintentional binding targets. These imaging FOVs and ROIs are saved so that, when the authors perform the SECOND run with the “dye of interest” they can compare the same locations and obtain a relative specificity between Cy3B (FIRST run) and the dye of interest (SECOND run).

My initial point is that if the above description of the experiment is in any way inaccurate the authors should consider rewriting the relevant sections to clarify any haziness or misconceptions which might obstruct the reader’s understanding. My follow-up (more significant) point is that I believe the value for Cy3B should not be exactly 1 with no variance between experiments (no error bars, as reported in Figure 3j). To clarify, I believe for every “dye of interest” the authors performed TWO runs (FIRST with Cy3B, SECOND with dye of interest). However, I believe the “relative specificity” value of 1 for Cy3B comes from the fact that the authors performed the FIRST run (reference run) with Cy3B, but did not perform a SECOND run (also with Cy3B) to compare to the FIRST run. Since there was only a single experiment performed (FIRST run), comparing the FIRST run to the FIRST run results in a value of 1 every time with no variance (hence a value of exactly 1 with no error bars for Cy3B, Figure 3j). However, the proper way to establish a base-line variance for this experiment would be to perform TWO runs with Cy3B, FIRST a relative run with Cy3B, then a wash, then a SECOND run on the same sample with the dye of interest being Cy3B. This should result in some amount of variance between the two runs which shows how consistently the method of measuring “relative specificity” behaves. Variance between the runs (FIRST and SECOND) across repeat experiments (triplicate repetitions at least) should result in a Cy3B point in Figure 3j that is not necessarily at 1 (though I suspect it will be close), and some amount of error bar from variance between experiments. If the authors did perform these two runs with Cy3B they should clarify, though I suspect they only performed the FIRST run for Cy3B, since they stated that “The relative specificity for all dyes compared to Cy3B, which is by definition equal to one...”. If the authors have not performed this two-run experiment on Cy3B they should perform this experiment and include this point with error bars in Figure 3j. Lastly, I will note that, though this difference seems pedantic, the difference of the Cy3B point from 1, and the error bars associated with it, is the most important part of Figure 3j, and is currently missing. This measurement would show the reader how confidently they can trust all other points and error bars reported in Figure 3j, as all these points are proportional to a Cy3B measurement and thus depend on how consistently Cy3B performs from measurement to measurement.

Minor Critiques:

- "DNA-PAINT images are rendered using Gaussians weighted by the CRLB localization precision as calculated according to Mortensen et al". Please define CRLB as it is not explicitly defined in this paper or in the Mortensen paper cited.
- Figure 1a: When comparing the "Bright & low bg" to "Bright & high bg" intensity timetrace, shouldn't the total intensity for the high background be higher? As written, the "Bright & high bg" is actually dimmer than the "Bright & low bg", as it has fewer total signal photons. Thus, in the resulting 2d images shown below it is hard to tell if the larger spots for the second image are a result of higher background or the signal being lower (a convolution already shown by the third image, "dim and high bg").
- Figure 2e-g: The y-axis "Locs per 20 s per d site" is confusing and should be described in the caption or the main text referencing this figure.
- "Of the dye's the photon output" typo in Extended Data Fig. 1 caption.
- The "N" statistic is not apparent across all Figures. For example, in Figure 2e-f it is unclear whether N represents number of origami, number of events, or number of 100 msec timestamps (I believe it is the latter). N values for the all values should be reported somewhere (e.g. "N=20,000 100 msec timestamps from N=1000 origami across N=3 experimental runs"). Similar counts should be included across all figures.
- Figure 3 caption: "Error bars represent s.d. across repeat experiments." should also report the number of experiments which contribute to the error bars.

Suggestions:

- Figure 1a: The green glow of the dye in the diagram makes it difficult to see what is going on. I would suggest that the dye was made to look more like Extended Data Fig. 1d.
- Extended Data Figure 1d: It might be worth mentioning somewhere that there are multiple "docking sites" within each "docking strand". E.g. the R1 Imager "AGGAGGA" can bind to multiple locations within the docking sequence "TCCTCCTCCTCCTCCT". Something which might not be apparent to readers of all levels.
- Spectral multiplexing in neurons: The authors state: "Cross-talk (the ratio of the number of localizations detected in the "incorrect" channel over the number of localizations detected in the "correct" channel) was below 1% for all cases.". I wish this information was reported across all dyes within the library that the authors measured. Such information might be useful for people selecting various groups of dyes from the ones available to them. This parameter could be obtained by performing the same experiment, but also performing off-color excitation (for example, if measuring a red dye also imaging in the blue/green channels). To ask the authors to repeat so many experiments to obtain this additional information is a ridiculous request and not at all required, but if the authors had access to such data already it would be informative for readers if they included it in the SI.
- Unrelated to this paper, but possibly helpful for future experiments in the authors' labs: I noticed in all Supplementary Figure f's that there was inconsistent improvement from PCA/PCD addition in the dyes' brightness (Photons per 100 ms) and photostability (Bright time, which is also dependent on koff). I also noticed that in several cases PCA/PCD addition seemed to hurt these parameters. Specifically, for Atto647N (a dye I am familiar with) the stability and brightness was significantly reduced upon addition of PCA/PCD, an effect which is not consistent with my observations. Lastly, I noticed that your experiments used PCD from Sigma Aldrich (cat: P8279). In the past, I found that using PCD (or rPCO) from OYCUS (cat: 46852004) resulted in a significant improvement to single-molecule photostability over the Sigma Aldrich counterpart while having similar brightness counts (this observation was made from a control experiment that compared these two sources of PCD specifically). You might consider ordering a small PCD sample from OYCUS and running some

comparative controls against Sigma Aldrich to see if that improves O2 scavenging dye stability in your future experiments!

Reviewer #2:

Remarks to the Author:

The work "The DNA-PAINT palette: A comprehensive performance analysis of fluorescent dyes" by Jungmann and coworkers present a systematic evaluation of 18 dyes for their use in DNA-PAINT.

Despite dyes properties are less important in DNA-PAINT than in other super-resolution methods, it is still relevant to make the appropriate choice and this is not yet addressed in the literature. Moreover, with many new dyes entering the market this work is timing making it a useful reference for the whole community, especially scientists approaching PAINT for the first time. It would have been even better if the study would have involved multiple laboratories to make the study even more robust. The systematic nature of this study is the power of this work, supporting the standardization and broad adoption of the technique.

The paper is well-written, results clearly presented and figures well organized. The methodology is sound and based on the experience of the group in DNA-PAINT. Statistical treatment of the data is fine. References are a bit limited (actually the whole introduction is very short), I would discuss some examples of current use of DNA-PAINT and the impact of the choice of the dyes may have of these fields.

The findings are not fully surprising (Cy3B and Atto643 are already the most used dyes) but several new insights are also emerging, for example for 488 dyes and the combination of buffer and dyes. Two very important points, qPAINT and aspecific binding, are touched but not enough in my opinion and expanding this would significantly improve the paper. Moreover, some more discussions would be helpful (see points below) to make the paper a bit less "dry".

Overall I recommend the work for publication in Nature Methods after the following points are addressed.

Major Points:

- There is very little discussion on why certain dyes or certain groups of dyes perform better. Can the authors relate the performance to chemical structure and photophysical parameters? Why some green-excited dyes seem to neatly outperform all the others?

-

- qPAINT is mentioned but not analyzed in the paper. I think quantitative measurements are much more challenging than simple imaging and it is where the choice of the dye may have the biggest impact. Can the authors address what are the best dyes for qPAINT? This can be based on the presented measurements having origami and NPC a known number of target molecules. Ranking can be made comparing counting precision and accuracy.

- The "sticking" is probably one of the most important issues in PAINT. The proposed analysis seems a bit limited. I strongly suggest to expand this part of the work. How is the background originated (sticking of the label or the dye?). In which cellular structures is more prominent? (e.g. membrane,

cytosol, nucleus). Does it depend on the buffer? This is very important in the context of less express targets and quantitative measurements. More measurements and analysis would be beneficial.

- Are the choices of the dyes and the DNA sequence disconnected or there is a potential cooperation between the two? The authors can discuss potential matching of dye performance (e.g. brightness and SNR) with the hybridization parameters (kon and koff for example).

- The authors repeat often that the choice of the dyes is critical, but is really always true? While some choices seem to have an impact (e.g. Cy3B) in other colors the difference between dyes is less significant, especially in the 488 channel. Can the authors be more quantitative and precise in this discussion? What really matters?

- Are all target the same? Can the authors show few cellular targets with the same dye (one of the best and one of the less-performing, no need to run it for all 18) and show how the parameters change?

Minor points

Page 1: "However, this specific method of achieving single molecule blinking leads to distinct requirements for fluorescent dyes". It would be nice to extend this part with a clear comparison of the features needed for STORM, PALM, and DNA-PAINT. Now the paragraph is a bit limited (there are more requirements for STORM, duty cycle, brightness, photostability) and the needs for DNA PAINT are mentioned only later.

Many acronyms not defined (may be difficult for non-experts) e.g. CRLB in page 3

Reviewer #3:

Remarks to the Author:

Summary:

In this manuscript, Steen et al. systematically evaluate fluorophores for their performance in DNA-PAINT. A systematic investigation of fluorophore performance for DNA-PAINT has indeed been missing; therefore, this analysis may be useful to the field. Additionally, the rigorous quantification of dye performance here is a useful template for future benchmarking of DNA-PAINT experiments. This analysis demonstrates the efficacy of Cy3B and provides high-quality alternatives for 488nm (CF488A) and 640nm (Atto643) excitation. However, it is somewhat unclear what novelty this analysis provides. Cy3B is already a commonly employed fluorophore in DNA-PAINT. It remains unclear why one would choose to use these other fluorophores instead of simply performing exchange-PAINT with Cy3B. Despite this critique, the work presented here is rigorous, high quality, and likely to be useful to the super-resolution microscopy field. Additional discussion and experimental details should be added to the manuscript prior to publication.

Major critiques:

1. Cy3B is the most popular dye in DNA-PAINT. The excellent performance of this fluorophore is well-appreciated; thus, it is unclear how this analysis improves the DNA PAINT field. The identification of excellent red (Atto643/ Atto647N) and blue (CF488A) options for imaging is potentially impactful.

However, Exchange-PAINT potentially allows imaging of an unlimited number of targets with the optimal fluorophore (Cy3B). What is the utility of the knowledge gained here? Further discussion of the multi-colored approach pursued here relative to Exchange-PAINT could clarify why this analysis is necessary.

2. The authors describe 4 parameters that affect the quality of DNA PAINT: (1) brightness, (2) signal to background ratio (SBR), (3) photo-induced damage of the docking strand, and (4) unspecific binding. How can the authors unambiguously differentiate brightness from signal to background ratio? These two parameters should be intertwined unless a fluorophore exhibits turn-on behavior when interacting with dsDNA. Fluorophores not bound to origami still produce fluorescence while diffusing in solution, and this fluorescence contributes to the background signal. A bright fluorophore should thus increase background. The manuscript would benefit from discussing or demonstrating how brightness and SBR can be independently modulated.

3. Exposure time is a critical parameter for DNA-PAINT, but it is not addressed in this manuscript. Bright fluorophores should be able to tolerate lower exposure times, potentially increasing SBR. The manuscript and analysis presented here would benefit tremendously from a brief exploration of the effect of exposure time on SBR for the several of the best fluorophores (e.g., for Atto 643, Cy3B, and CF488A).

4. In supplementary table 1, the imager concentration is said to be 300-500pM for red dyes of interest. In images with U20S cells, imager concentrations similarly vary for blue, green, and red dyes. The SBR may be a function of imager concentration. The author should provide a plot of SBR as a function of imager concentration for different fluorophores, or at the very least indicate which fluorophore was used with which concentration.

5. Please provide data to substantiate the choice of excitation intensities for 488nm, 560nm, and 640nm lasers. Is the tradeoff of photoinduced damage at 488nm excitation so severe that low laser powers (and consequently low photon outputs) are necessary? The authors state "the achievable localization precision is primarily governed by high photon output and SBR (Fig. 2i), however binding site damage reduces overall image quality." If a higher laser power was used for 488 excitation the brightness and perhaps SBR should increase. Would that increase offset the increased damage of docking sites (which the authors intimate is a secondary concern)? Data is desperately needed to substantiate the author's decisions here.

6. It is unclear how the SBR is calculated. The authors state that "the background value is a per-pixel offset." It is unclear to me whether the description of the area integral is applied for the photon count or the calculation of the offset. Is this offset taken to be the dark signal of the camera? Is it taken to be the integrated intensity of a 7x7 pixel area in which no localization is currently present? Does this offset account for the higher fluorescence due to the presence of diffusing fluorophores and/or increased excitation intensity?

7. For main figures 2-4, it is unclear how many times the imaging was repeated.

8. In Figure 3, it is unclear how the authors define whether they are able to resolve the eight-fold symmetry of the nuclear pore. In the images show, between 8-9 clusters of localization could be present in the Cy3B image, and between 8 and 11 could be present in the Atto643 image. Did the authors utilize a clustering analysis to determine whether they could discern the eight-fold symmetry?

Minor Comments:

In figure 2i, the point spread function for Atto643 exhibits a red background, suggesting a photon count of about 1000-2000 photons. The backgrounds of Cy3B and CF488 are entirely black. Despite this visual difference, The SBR of Atto643 (4.7) is measured to be higher than CF488A (3.7). Is there background in the CF488A image that is not rendered by the colormap? Perhaps providing the average of the background intensity for each fluorophore in 2i would be helpful.

The manuscript would benefit from an SI figure depicting the calculation of docking site destruction, histogram of "r" ratios, and exponential fitting for several representative fluorophores as discussed in the methods. It is hard to understand the process as currently described.

I do not understand the units of the "stability" plots in figure 2 and in the SI figures 1-18. What is the meaning of "loss per 20 s per d. site." I am confused about the "d." I believe this could be "docking" site. Please clarify the meaning of this abbreviation.

In figure 3f, atto565 and CF568 exhibit superior SBR to Cy3B, but worse nearest neighbor precision. Can the authors comment on how this occurred?

Author Rebuttal to Initial comments

Reviewer #1:

Remarks to the Author:

In this manuscript the authors characterize a library of 18 different fluorescent dyes for use in DNA-PAINT experiments. The dyes are first analyzed on a well-behaved DNA origami system to obtain their fluorescent output and signal-background ratio (and the consequent spatial resolution), as well as the dyes' impact on site stability. All dyes were then tested in a cellular experiment imaging a Nup96 nuclear pore complex to obtain how these properties would persist in a more complicated environment, and to additionally quantify the dyes' propensity for (undesired) non-specific binding. Lastly, three of the best-performing dyes corresponding to the different excitation channels were selected for use in a multiplexing experiment to image neurons, verifying that high-resolution imaging could be performed with simultaneous measurements involving dye-mixtures. While the methodologies are now routine in the field (in fact the DNA paint calibration markers represent one of the first commercial applications of DNA origami structures), the fundamental science is sound, the presentation of the data is clear, and, most importantly, the utility of such a report as a resource for future DNA-PAINT experiments is certainly beneficial to the single-molecule and broader scientific community as a whole. I believe there are a few areas the current paper could be improved in order to become more accessible to the broader scientific community and to reinforce the results displayed in the figures which are parsed into "major" and "minor" critiques below. As Nature Methods is a journal which publishes "performance comparisons of related, established methods", this paper falls well within the boundaries of the journal, and I would recommend publication after addressing the critiques outlined below.

We thank the reviewer for their very kind remarks.

Major Critique 1: Quantification and discussion of dye-dependent docking site destruction

This critique comes in two parts: 1) the visual clarity of how docking-site stability was calculated and 2) a discussion of how important the docking-site stability parameter is compared to other dye-related parameters within DNA PAINT experiments.

First (1), it is difficult to visualize how the docking-site stability reported (as in Figure 2 e-g, Table 1, etc.) is calculated. The methods section states that "The ratio $r = t_{end}/t_d$ was calculated for each docking site and all ratios plotted as a histogram.", however these histograms (and their corresponding exponential fits) are not shown, and neither are the relevant μ values and their related cutoffs. Since the authors have access to this data I believe it should be reported as sub-figures in the Supporting Information, which would aid the reader in visualizing how well the r-histograms fit the exponential decay, how strict the 4μ cutoff is, etc. Currently the reader only has access to the "final result" as reported in Table 1.

We agree that the previous explanation was not fully clear, and have now added Extended Data Fig. 5, which features a graphical representation of how r is calculated for each trace, how plotting r for all traces in a measurement yields the exponential decay, as well as sample traces, histograms, and μ values for various dyes.

Additionally, the y-axis describing "locs per 20 s per d. site" (in Fig 2 e-g and all SI figures) is not described, and the relation between "Localization drop" and "Destroyed binding sites" is not defined or explored.

We thank the reviewer for raising this point. We have now added clarifying statements to the figure captions. Extended Data Fig. 4 now shows the percentage of destroyed docking sites plotted against the drop in localizations, illustrating the correlation.

It is not intuitive to me that some dyes would result in binding sites being destroyed (Cy5, Cy3b, etc.) while showing fewer (or no) "Localization drop" (would binding-site destruction not cause localizations to drop?). I assume that this discrepancy is due simply to statistical variance, yet the differences between the "Destroyed sites" and "Localization drop" values at the end of Table 1 are, in some cases, well outside the variance reported, and as the intermediate data is not presented in any other places it is difficult to see why that is the case. I would like to see the intermediate data (r-histograms, fits, μ values, etc.) presented somewhere, and for the authors to include a short discussion in either the methods section or the SI describing these parameters, as they are relevant to the binding-site stability portion of the paper.

We thank the reviewer for raising this point. We now provide intermediate data in Extended Data Figure 5 and show the correlation between decrease in localizations and destruction of docking sites in Extended Data Figure 4. We agree that discrepancies are most likely due to statistical variance.

Second (2) it seems to me a large percentage of the target-readers for this paper would be labs that are "beginners" at DNA-PAINT and are selecting potential dyes for use in their future experiments. In other words, the readers are labs which might not know how significant a 4.0% loss of binding sites is over 2,000 seconds compared to a precision difference of 2 nm at 70 mW excitation power. I believe the accessibility of this paper would be significantly improved by a brief discussion about the relative importance of these various dye-dependent parameters, in addition to the (already included) selection by the authors of the "best-performing dyes".

This is a very important point; We have now added a paragraph explaining how consistent sampling (e.g. prevention of loss/destruction of binding sites) enables faithful analysis by e.g. qPAINT. Furthermore, we investigated the effect of higher laser excitation power on the destruction of binding sites and localization precision (Extended Data Figure 6) and would generally recommend to use the lowest laser excitation power necessary to achieve a given target localization precision in order to minimize detrimental effects on docking site destruction.

Major Critique 2: Reporting of the "relative dye specificity in an individual measurement"

It was not 100% clear to me exactly how the "dye specificity in an individual measurement" was calculated, so allow me to describe what I think the authors did to calculate this parameter: For each "dye of interest" the authors ran TWO sequential DNA-PAINT experiments (runs), FIRST a run with Cy3B as a reference, then a wash, then a SECOND run with the "dye of interest". From the nuclear images of the FIRST run (as represented top of Figure 3g-i), the authors manually selected several ROIs describing Nup96s before algorithmically finding all such ROIs, they then manually reviewed these ROIs to ensure accuracy. The authors also moved the FOV to image the cell away from the nucleus which should contain no Nup96, but might contain some unintentional binding targets. These imaging FOVs and ROIs are saved so that, when the authors perform the SECOND run with the "dye of interest" they can compare the same locations and obtain a relative specificity between Cy3B (FIRST run) and the dye of interest (SECOND run).

This is a correct summary of the process. However, we see that this may be confusing, so we have added Extended Data Fig. 8 to clarify this process for the reader.

My initial point is that if the above description of the experiment is in any way inaccurate the authors should consider rewriting the relevant sections to clarify any haziness or misconceptions which might obstruct the reader's understanding. My follow-up (more significant) point is that I believe the value for Cy3B should not be exactly 1 with no variance between experiments (no error bars, as reported in Figure 3j). To clarify, I believe for every "dye of interest" the authors performed TWO runs (FIRST with Cy3B, SECOND with dye of interest). However, I believe the "relative specificity" value of 1 for Cy3B comes from the fact that the authors performed the FIRST run (reference run) with Cy3B, but did not perform a SECOND run (also with Cy3B) to compare to the FIRST run. Since there was only a single experiment performed (FIRST run), comparing the FIRST run to the FIRST run results in a value of 1 every time with no variance (hence a value of exactly 1 with no error bars for Cy3B, Figure 3j). However, the proper way to establish a base-line variance for this experiment would be to perform TWO runs with Cy3B, FIRST a relative run with Cy3B, then a wash, then a SECOND run on the same sample with the dye of interest being Cy3B. This should result in some amount of variance between the two runs which shows how consistently the method of measuring "relative specificity" behaves. Variance between the runs (FIRST and SECOND) across repeat experiments (triplicate repetitions at least) should result in a Cy3B point in Figure 3j that is not necessarily at 1 (though I suspect it will be close), and some amount of error bar from variance between experiments. If the authors did perform these two runs with Cy3B they should clarify, though I suspect they only performed the FIRST run for Cy3B, since they stated that "The relative specificity for all dyes compared to Cy3B, which is by definition equal to one...". If the authors have not performed this two-run experiment on Cy3B they should perform this experiment and include this point with error bars in Figure 3j. Lastly, I will note that, though this difference seems pedantic, the difference of the Cy3B point from 1, and the error bars associated with it, is the most important part of Figure 3j, and is currently missing. This measurement would show the reader how confidently they can trust all other points and error bars reported in Figure 3j, as all these points are proportional to a Cy3B measurement and thus depend on how consistently Cy3B performs from measurement to measurement.

This is a valid and important point. The reviewer is correct in that we initially did not plot the variance between the first and second run of Cy3B. We, however, have now performed experiments in which both the reference dye and dye of interest are Cy3B (to ensure the validity of this approach), and have now updated the plot to include relative specificity of Cy3B (which is 1.03 ± 0.09).

Minor Critiques:

- "DNA-PAINT images are rendered using Gaussians weighted by the CRLB localization precision as calculated according to Mortensen et al". Please define CRLB as it is not explicitly defined in this paper or in the Mortensen paper cited.

We thank the reviewer for pointing this out. We have now added a section in the methods referencing the formula used for weighing the Gaussian rendering and have removed the mentioning of CRLB.

- Figure 1a: When comparing the “Bright & low bg” to “Bright & high bg” intensity timetrace, shouldn’t the total intensity for the high background be higher? As written, the “Bright & high bg” is actually dimmer than the “Bright & low bg”, as it has fewer total signal photons. Thus, in the resulting 2d images shown below it is hard to tell if the larger spots for the second image are a result of higher background or the signal being lower (a convolution already shown by the third image, “dim and high bg”).

We are grateful to the reviewer for pointing this out. This is correct, the figure has been changed accordingly.

- Figure 2e-g: The y-axis “Locs per 20 s per d site” is confusing and should be described in the caption or the main text referencing this figure.

Thank you for pointing this out, we now clarify this in the figure captions

- “Of the dye’s the photon output” typo in Extended Data Fig. 1 caption.

Thank you, this has been corrected.

- The “N” statistic is not apparent across all Figures. For example, in Figure 2e-f it is unclear whether N represents number of origami, number of events, or number of 100 msec timestamps (I believe it is the latter). N values for the all values should be reported somewhere (e.g. “N=20,000 100 msec timestamps from N=1000 origami across N=3 experimental runs”). Similar counts should be included across all figures.

The number of repeat experiments has been added to each figure where applicable. In figure 2e-g the \bar{N} value refers to the mean number of photons per localization. We have clarified this in the figure caption.

- Figure 3 caption: “Error bars represent s.d. across repeat experiments.” should also report the number of experiments which contribute to the error bars.

The number of repeat experiments has been added to each figure where applicable.

Suggestions:

- Figure 1a: The green glow of the dye in the diagram makes it difficult to see what is going on. I would suggest that the dye was made to look more like Extended Data Fig. 1d.

We thank the reviewer for this suggestion, which we have implemented.

- Extended Data Figure 1d: It might be worth mentioning somewhere that there are multiple “docking sites” within each “docking strand”. E.g. the R1 Imager “AGGAGGA” can bind to multiple locations within the docking sequence “TCCTCCTCCTCCTCCT”. Something which might not be apparent to readers of all levels.

This is a very good point; we have clarified this in the discussion of docking site damage / destruction.

- Spectral multiplexing in neurons: The authors state: "Cross-talk (the ratio of the number of localizations detected in the "incorrect" channel over the number of localizations detected in the "correct" channel) was below 1% for all cases." I wish this information was reported across all dyes within the library that the authors measured. Such information might be useful for people selecting various groups of dyes from the ones available to them. This parameter could be obtained by performing the same experiment, but also performing off-color excitation (for example, if measuring a red dye also imaging in the blue/green channels). To ask the authors to repeat so many experiments to obtain this additional information is a ridiculous request and not at all required, but if the authors had access to such data already it would be informative for readers if they included it in the SI.

We are sorry, but this data does not exist. We performed such experiments using DNA origami for the three chosen dyes (thus yielding the "below 1%"), but not for all 18 dyes in this paper.

- Unrelated to this paper, but possibly helpful for future experiments in the authors' labs: I noticed in all Supplementary Figure f's that there was inconsistent improvement from PCA/PCD addition in the dyes' brightness (Photons per 100 ms) and photostability (Bright time, which is also dependent on koff). I also noticed that in several cases PCA/PCD addition seemed to hurt these parameters. Specifically, for Atto647N (a dye I am familiar with) the stability and brightness was significantly reduced upon addition of PCA/PCD, an effect which is not consistent with my observations. Lastly, I noticed that your experiments used PCD from Sigma Aldrich (cat: P8279). In the past, I found that using PCD (or rPCO) from OYCUS (cat: 46852004) resulted in a significant improvement to single-molecule photostability over the Sigma Aldrich counterpart while having similar brightness counts (this observation was made from a control experiment that compared these two sources of PCD specifically). You might consider ordering a small PCD sample from OYCUS and running some comparative controls against Sigma Aldrich to see if that improves O2 scavenging dye stability in your future experiments!

We are grateful for this advice and will evaluate these reagents in future experiments.

Reviewer #2:

Remarks to the Author:

The work "The DNA-PAINT palette: A comprehensive performance analysis of fluorescent dyes" by Jungmann and coworkers present a systematic evaluation of 18 dyes for their use in DNA-PAINT.

Despite dyes properties are less important in DNA-PAINT than in other super-resolution methods, it is still relevant to make the appropriate choice and this is not yet addressed in the literature. Moreover, with many new dyes entering the market this work is timing making it a useful reference for the whole community, especially scientists approaching PAINT for the first time. It would have been even better if the study would have involved multiple laboratories to make the study even more robust. The systematic nature of this study is the power of this work, supporting the standardization and broad adoption of the technique.

We thank the reviewer for these kind comments and indeed hope that these evaluation metrics will be widely adopted by the community.

The paper is well-written, results clearly presented and figures well organized. The methodology is sound and based on the experience of the group in DNA-PAINT. Statistical treatment of the data is fine. References are a bit limited (actually the whole introduction is very short), I would discuss some examples of current use of DNA-PAINT and the impact of the choice of the dyes may have of these fields.

We are grateful for this suggestion. We have expanded the introduction accordingly to discuss these points.

The findings are not fully surprising (Cy3B and Atto643 are already the most used dyes) but several new insights are also emerging, for example for 488 dyes and the combination of buffer and dyes. Two very important points, qPAINT and aspecific binding, are touched but not enough in my opinion and expanding this would significantly improve the paper. Moreover, some more discussions would be helpful (see points below) to make the paper a bit less "dry".

Overall I recommend the work for publication in Nature Methods after the following points are addressed.

Major Points:

- There is very little discussion on why certain dyes or certain groups of dyes perform better. Can the authors relate the performance to chemical structure and photophysical parameters? Why some green-excited dyes seem to neatly outperform all the others?

We thank the reviewer for raising this point. While the performance differences of many dyes in our study can be attributed to intrinsic photophysical properties of the dyes such as different quantum yields, the higher signal-to-background ratio of Cy3b, Atto565, and CF568 in DNA-PAINT experiments cannot be directly linked to such properties. The differences might be related to somewhat different interactions of the dye molecules with DNA strands or changes in single-stranded to double-stranded transition. However, these remain speculative and are subject to further photophysical studies, which are out of the scope of this report.

- qPAINT is mentioned but not analyzed in the paper. I think quantitative measurements are much more challenging than simple imaging and it is where the choice of the dye may have the biggest impact. Can the authors address what are the best dyes for qPAINT? This can be based on the presented measurements having origami and NPC a known number of target molecules. Ranking can be made comparing counting precision and accuracy.

We thank the reviewer for pointing this out. In general, the requirement for qPAINT is consistent

sampling with no reduction in number of localizations over time and no destruction of docking sites. We have added a sentence, highlighting this specific requirement for accurate qPAINT quantification

- The “sticking” is probably one of the most important issues in PAINT. The proposed analysis seem a bit limited. I strongly suggest to expand this part of the work. How is the background originated (sticking of the label or the dye?). In which cellular structures is more prominent? (e.g. membrane, cytosol, nucleus). Does it depend on the buffer? This is very important in the context of less express targets and quantitative measurements. More measurements and analysis would be beneficial.

We thank the reviewer for raising this point. We have now tested a subset of dyes on a different target, namely the mitochondrial membrane-associated protein Tom20, and could show that the relative “stickiness” is consistent with the NPC measurements (see Extended Data Fig. 9).

- Are the choices of the dyes and the DNA sequence disconnected or there is a potential cooperation between the two? The authors can discuss potential matching of dye performance (e.g. brightness and SNR) with the hybridization parameters (kon and koff for example).

We are grateful for this suggestion. We have now added an investigation of three sequences with three dyes (Extended Data Fig. 7). Overall, the performances are very similar between sequences, the most significant differences are the bright times.

- The authors repeat often that the choice of the dyes is critical, but is really always true? While some choices seem to have an impact (e.g. Cy3B) in other colors the difference between dyes is less significant, especially in the 488 channel. Can the authors be more quantitative and precise in this discussion? What really matters?

We thank the reviewer for raising this potentially confusing issue. While we agree, that Cy3B seems to be the most sensible choice in a single-plex or Exchange-PAINT experiment, there are instances where another dye choice is necessary. We have now added a clarification paragraph in the introduction describing these instances. However, we respectfully disagree that the choice of dye for other wavelength regimes is less significant. Even though the absolute performance of e.g. 488 nm excitation dyes compared to Cy3B is lower, it is still important to choose the best-performing dye for each spectral range.

- Are all targets the same? Can the authors show few cellular targets with the same dye (one of the best and one of the less-performing, no need to run it for all 18) and show how the parameters change?

We thank the reviewer for this suggestion. While the multiplexing experiment in the neurons (Figure 4) already broadly recapitulates the performance differences between different dyes, we have now added a quantitative comparison of select dyes targeting the mitochondrial membrane-associated protein Tom20. We see results that are very consistent with those from NPC measurements.

Minor points

Page 1: “However, this specific method of achieving single molecule blinking leads to distinct requirements for fluorescent dyes”. It would be nice to extend this part with a clear comparison of the features needed for STORM, PALM, and DNA-PAINT. Now the paragraph is a bit limited (there are more requirements for STORM, duty cycle, brightness, photostability) and the needs for DNA PAINT are mentioned only later.

While we agree with the reviewer regarding specific requirements for STORM, PALM, and PAINT, we believe it is most relevant to focus on the requirements for DNA-PAINT in our current study. The requirements for STORM are thoroughly discussed by Dempsey et al. in their 2011 Nature Methods paper, which we point to (reference 10).

Many acronyms not defined (may be difficult for non-experts) e.g. CRLB in page 3

We apologize for this and have now carefully reviewed and defined all acronyms.

Reviewer #3:

Remarks to the Author:

Summary:

In this manuscript, Steen et al. systematically evaluate fluorophores for their performance in DNA-PAINT. A systematic investigation of fluorophore performance for DNA-PAINT has indeed been missing; therefore, this analysis may be useful to the field. Additionally, the rigorous quantification of dye performance here is a useful template for future benchmarking of DNA-PAINT experiments. This analysis demonstrates the efficacy of Cy3B and provides high-quality alternatives for 488nm (CF488A) and 640nm (Atto643) excitation. However, it is somewhat unclear what novelty this analysis provides. Cy3B is already a commonly employed fluorophore in DNA-PAINT. It remains unclear why one would choose to use these other fluorophores instead of simply performing exchange-PAINT with Cy3B. Despite this critique, the work presented here is rigorous, high quality, and likely to be useful to the super-resolution microscopy field. Additional discussion and experimental details should be added to the manuscript prior to publication.

We are grateful for the positive review of our work by this referee. We agree that it might not have been sufficiently motivated in the original submission why one would not simply use Cy3B in an Exchange-PAINT-type experiment. To address this, we have now added a section discussing these use cases. Furthermore, we think this quantitative analysis pipeline will prove very useful if and when dyes that improve upon Cy3B become available.

Major critiques:

1. Cy3B is the most popular dye in DNA-PAINT. The excellent performance of this fluorophore is well-appreciated; thus, it is unclear how this analysis improves the DNA PAINT field. The identification of excellent red (Atto643/ Atto647N) and blue (CF488A) options for imaging is potentially impactful. However, Exchange-PAINT potentially allows imaging of an unlimited number of targets with the optimal

fluorophore (Cy3B). What is the utility of the knowledge gained here? Further discussion of the multi-colored approach pursued here relative to Exchange-PAINT could clarify why this analysis is necessary.

We do agree that this has not been motivated sufficiently. We have now addressed this by adding the following paragraph to the introduction: “While it is broadly understood that the fluorescent dye Cy3B offers excellent performance, there are instances in which either using Cy3B or Exchange-PAINT is not possible. The former occurs when using samples tagged with Red Fluorescent Protein (RFP) or similar markers featuring spectra that overlap with that of Cy3B or imaging on microscope setups that do not feature a sufficiently powerful 560 nm laser line. The latter is the case when imaging samples with very slow diffusion, such as dense tissues, or non-adherent cells that risk being washed away during buffer exchange. Especially in these cases, spectral multiplexing without buffer exchange is an important technique to enable imaging multiple targets.”

2. The authors describe 4 parameters that affect the quality of DNA PAINT: (1) brightness, (2) signal to background ratio (SBR), (3) photo-induced damage of the docking strand, and (4) unspecific binding. How can the authors unambiguously differentiate brightness from signal to background ratio? These two parameters should be intertwined unless a fluorophore exhibits turn-on behavior when interacting with dsDNA. Fluorophores not bound to origami still produce fluorescence while diffusing in solution, and this fluorescence contributes to the background signal. A bright fluorophore should thus increase background. The manuscript would benefit from discussing or demonstrating how brightness and SBR can be independently modulated.

We do agree that if the dye would be unaffected by potential differences between bound vs. unbound states of the DNA-PAINT imager strand, the signal-to-background ratios should be similar for different dyes. However, we do observe differences in the signal-to-background for certain dyes. The differences might be related to somewhat different interactions of the dye molecules with DNA strands or changes in single-stranded to double-stranded transition (e.g. quenching of dyes for imager strands not bound to the docking strand, see e.g. Figure S3 in <https://doi.org/10.1021/nl103427w>). However, these remain mostly speculative and are subject to further photophysical studies, which are out of the scope of this report. To further clarify the difference between brightness and signal-to-background, we have now added Extended Data Fig. 3, detailing how these properties are calculated and how they can be independently measured.

3. Exposure time is a critical parameter for DNA-PAINT, but it is not addressed in this manuscript. Bright fluorophores should be able to tolerate lower exposure times, potentially increasing SBR. The manuscript and analysis presented here would benefit tremendously from a brief exploration of the effect of exposure time on SBR for the several of the best fluorophores (e.g., for Atto 643, Cy3B, and CF488A).

We thank the reviewer for raising this issue. However, modulating the exposure time does not alter SBR per our SBR definition. We note that we only consider what we call “center frames” (see materials and methods) in our calculation and analysis, which ensures that an imager strand is always fully bound throughout the frame exposure. In this case, both signal photons and background photons are proportional to the exposure time and thus cancel out in the SBR calculation.

4. In supplementary table 1, the imager concentration is said to be 300-500pM for red dyes of interest. In images with U2OS cells, imager concentrations similarly vary for blue, green, and red dyes. The SBR may be a function of imager concentration. The author should provide a plot of SBR as a function of imager concentration for different fluorophores, or at the very least indicate which fluorophore was used with which concentration.

We thank the reviewer for raising this point. We agree that SBR is influenced by imager concentrations. As proposed by the referee, we now report the exact concentration used for each experiment in Supplementary Table 1.

5. Please provide data to substantiate the choice of excitation intensities for 488nm, 560nm, and 640nm lasers. Is the tradeoff of photoinduced damage at 488nm excitation so severe that low laser powers (and consequently low photon outputs) are necessary? The authors state “the achievable localization precision is primarily governed by high photon output and SBR (Fig. 2i), however binding site damage reduces overall image quality.” If a higher laser power was used for 488 excitation the brightness and perhaps SBR should increase. Would that increase offset the increased damage of docking sites (which the authors intimate is a secondary concern)? Data is desperately needed to substantiate the author’s decisions here.

We are grateful for the suggestion of the reviewer and now provide the additional Extended Data Fig. 6 to show the influence of excitation power on localization precision, localization drop, and binding site destruction. Overall, we chose an approach that balanced resolution vs. sample damage.

6. It is unclear how the SBR is calculated. The authors state that “the background value is a per-pixel offset.” It is unclear to me whether the description of the area integral is applied for the photon count or the calculation of the offset. Is this offset taken to be the dark signal of the camera? Is it taken to be the integrated intensity of a 7x7 pixel area in which no localization is currently present? Does this offset account for the higher fluorescence due to the presence of diffusing fluorophores and/or increased excitation intensity?

We apologize for the potential confusion regarding the SBR calculation. We have now added Extended Data Fig. 3 to clarify photon signal, background and SBR calculation.

7. For main figures 2-4, it is unclear how many times the imaging was repeated.

All experiments were conducted in triplicate. This has now been clarified in the figure legends.

8. In Figure 3, it is unclear how the authors define whether they are able to resolve the eight-fold symmetry of the nuclear pore. In the images show, between 8-9 clusters of localization could be present in the Cy3B image, and between 8 and 11 could be present in the Atto643 image. Did the authors utilize a clustering analysis to determine whether they could discern the eight-fold symmetry?

We thank the reviewer for pointing us to this issue. We have now removed the statement regarding resolving the 8-fold symmetry of the NPC, as this is not relevant in our analysis procedure for the current study.

Minor Comments:

In figure 2i, the point spread function for Atto643 exhibits a red background, suggesting a photon count of about 1000-2000 photons. The backgrounds of Cy3B and CF488 are entirely black. Despite this visual difference, The SBR of Atto643 (4.7) is measured to be higher than CF488A (3.7). Is there background in the CF488A image that is not rendered by the colormap? Perhaps providing the average of the background intensity for each fluorophore in 2i would be helpful.

The color map and scale were chosen to be identical for all three dyes to highlight the absolute rather than relative differences.

The manuscript would benefit from an SI figure depicting the calculation of docking site destruction, histogram of “r” ratios, and exponential fitting for several representative fluorophores as discussed in the methods. It is hard to understand the process as currently described.

We are thankful to the reviewer for bringing up this issue. This is a very important point; Extended Data Fig. 5 has been added for clarification.

I do not understand the units of the “stability” plots in figure 2 and in the SI figures 1-18. What is the meaning of “loss per 20 s per d. site.” I am confused about the “d.” I believe this could be “docking” site. Please clarify the meaning of this abbreviation.

An explanation for this was indeed missing from the figure legend, we do apologize for this oversight. A clarification, namely that this refers to “localizations per 20 seconds per docking site over time” has been added.

In figure 3f, atto565 and CF568 exhibit superior SBR to Cy3B, but worse nearest neighbor precision. Can the authors comment on how this occurred?

This is due to the absolute photon output, which was higher for Cy3B.

Decision Letter, first revision:

Dear Ralf,

Thank you for submitting your revised manuscript "The DNA-PAINT palette: A comprehensive performance analysis of fluorescent dyes" (N METH-AS54444A). It has now been seen by the original referees and their comments are below. The reviewers find that the paper has improved in revision, and therefore we'll be happy in principle to publish it in Nature Methods, pending minor revisions to comply with our editorial and formatting guidelines.

TRANSPARENT PEER REVIEW

Please note: we allow redactions to authors' rebuttal and reviewer comments in the interest of confidentiality. If you are concerned about the release of confidential data, please let us know specifically what information you would like to have removed. Please note that we cannot incorporate redactions for any other reasons. Reviewer names will be published in the peer review files if the reviewer signed the comments to authors, or if reviewers explicitly agree to release their name. For more information, please refer to our FAQ page.

ORCID

Sincerely,
Rita

Rita Strack, Ph.D.
Senior Editor
Nature Methods

Reviewer #1 (Remarks to the Author):

I commend the authors for their work addressing reviewer's comments. The authors have been thorough on their reporting of statistics (N=...), performed an additional experiment quantifying Cy3b's specificity (Figure 3j), and have included a significant amount of additional data (including the much-requested "intermediate data") as shown in the addition of extended data. I am particularly impressed with their work on Extended Data 5, as it visualizes a non-intuitive analysis rather elegantly. I would recommend a (very minor) text adjustments to help point readers to the Extended Data before final publication.

In 5. Docking site destruction: "... so the histogram of all r values also follows an exponential distribution (see Extended Data 5).

Reviewer #2 (Remarks to the Author):

The authors addressed most of my points and I recommend the manuscript for publication

Reviewer #3 (Remarks to the Author):

The revised manuscript from Steen et al. provides significant additional information regarding their analysis of the effects of dye choice on DNA PAINT. In particular, the inclusion of Extended Data Figures 3-6 significantly improves my understanding of the authors' analysis pipeline.

I have no further comments for the authors. I believe many of these metrics will be useful in my own research, and I heartily recommend publication.

Reviewer #3 (Remarks on code availability):

I was not able to access the code. Perhaps I simply don't know how to properly use the doi that has been provided.

Author Rebuttal, first revision:**Reviewer #1:**

Remarks to the Author:

I commend the authors for their work addressing reviewer's comments. The authors have been thorough on their reporting of statistics (N=...), performed an additional experiment quantifying Cy3b's specificity (Figure 3j), and have included a significant amount of additional data (including the much-requested "intermediate data") as shown in the addition of extended data. I am particularly impressed with their work on Extended Data 5, as it visualizes a non-intuitive analysis rather elegantly. I would recommend a (very minor) text

adjustments to help point readers to the Extended Data before final publication.

In 5. Docking site destruction: "... so the histogram of all r values also follows an exponential distribution (see Extended Data 5).

We thank the reviewer for their very kind remarks, and have added a reference to Extended Data Fig. 5 in the Methods as suggested.

Reviewer #2:

Remarks to the Author:

The authors addressed most of my points and I recommend the manuscript for publication

We appreciate the reviewer's kind comments.

Reviewer #3:

Remarks to the Author:

The revised manuscript from Steen et al. provides significant additional information regarding their analysis of the effects of dye choice on DNA PAINT. In particular, the inclusion of Extended Data Figures 3-6 significantly improves my understanding of the authors' analysis pipeline.

I have no further comments for the authors. I believe many of these metrics will be useful in my own research, and I heartily recommend publication.

We are grateful to the reviewer for their remarks and look forward to the application of the metrics we propose.

Final Decision Letter:

Dear Ralf,

I am pleased to inform you that your Analysis, "The DNA-PAINT palette: A comprehensive performance analysis of fluorescent dyes", has now been accepted for publication in Nature Methods. The received and accepted dates will be Nov 14, 2023 and June 21, 2024. This note is intended to let you know what to expect from us over the next month or so, and to let you know where to address

any further questions.

Over the next few weeks, your paper will be copyedited to ensure that it conforms to Nature Methods style. Once your paper is typeset, you will receive an email with a link to choose the appropriate publishing options for your paper and our Author Services team will be in touch regarding any additional information that may be required. It is extremely important that you let us know now whether you will be difficult to contact over the next month. If this is the case, we ask that you send us the contact information (email, phone and fax) of someone who will be able to check the proofs and deal with any last-minute problems.

Please note that *Nature Methods* is a Transformative Journal (TJ). Authors may publish their research with us through the traditional subscription access route or make their paper immediately open access through payment of an article-processing charge (APC). Authors will not be required to make a final decision about access to their article until it has been accepted. Find out more about Transformative Journals

If you are active on Twitter/X, please e-mail me your and your coauthors' handles so that we may tag you when the paper is published.

Best regards,
Rita

Rita Strack, Ph.D.
Senior Editor
Nature Methods